# Feasibility of reconstructing the summer basin–scale sea surface partial pressure of carbon dioxide from sparse in situ observations over the South China Sea

Guizhi Wang[1,2], Samuel S. P. Shen[3], Yao Chen[3], Yan Bai[4], Huan Qin[3], Zhixuan Wang[1], Baoshan Chen[1], Xianghui Guo[1], Minhan Dai[1]

[1]State Key Laboratory of Marine Environmental Science and College of Ocean and Earth Sciences, Xiamen University, Xiamen, 361102, China
[2]Fujian Provincial Key Laboratory for Coastal Ecology and Environmental Studies, Xiamen University, Xiamen, 361102, China
[3]Department of Mathematics and Statistics, San Diego State University, San Diego, CA 92182, USA
[4]State Key Laboratory of Satellite Ocean Environment Dynamics, Second Institute of Oceanography, State Oceanic Administration, Hangzhou, 310012, China

*Correspondence to*: Guizhi Wang (gzhwang@xmu.edu.cn), Samuel S. P. Shen (sshen@sdsu.edu)

**Abstract.** Sea surface partial pressure of $CO_2$ ($pCO_2$) data with a high spatial-temporal resolution are important in studying the global carbon cycle and assessing the oceanic carbon uptake. However, the observed sea surface $pCO_2$ data are usually limited in spatial and temporal coverage, especially in marginal seas. This study provides an approach to reconstruct the complete sea surface $pCO_2$ field in the South China Sea (SCS) with a grid resolution of 0.5°×0.5° over the period of 2000–2017 using both remote-sensing derived $pCO_2$ and observed underway $pCO_2$, among which the gridded underway $pCO_2$ data in 2004, 2005, and 2006 are presented for the first time. Empirical orthogonal functions (EOFs) were computed from the remote sensing derived $pCO_2$. Then, a multilinear regression was applied to the observed $pCO_2$ as the response variable with the EOFs as the explanatory variables. EOF1 explains the general spatial pattern of $pCO_2$ in the SCS. EOF2 shows the pattern influenced by the Pearl River plume on the northern shelf and slope. EOF3 is consistent with the pattern influenced by coastal upwelling along the north coast of the SCS. When $pCO_2$ observations cover a sufficiently large area, the reconstructed fields successfully display a pattern of relatively high $pCO_2$ in the mid-and-southern basin. The rate of sea surface $pCO_2$ increase in the SCS is 2.4±0.8 µatm per year based on the spatial average of the reconstructed $pCO_2$ over the period of 2000–2017. This is consistent with the temporal trends at Station SEATS (18º N, 116º E) in the northern basin of the SCS and at Station HOT (22º45´ N, 158º W) in the North Pacific. We validated our reconstruction with a leave-one-out cross-validation approach, which yields the root-mean-square error (RMSE) in the range of 2.4–5.2 µatm, smaller than the spatial standard deviation of our reconstructed data and much smaller than the spatial standard deviation of the observed underway data. The RMSE between the reconstructed summer $pCO_2$ and the observed underway $pCO_2$ is no larger than 31.7 µatm, in contrast to (a) the RMSE from 12.8–89.0 µatm between the remote-sensing derived $pCO_2$ and the underway data, and (b) the RMSE from 32.6–44.5 µatm between the neural network produced $pCO_2$ and the underway data. The difference between the reconstructed $pCO_2$ and

those calculated from observations at Station SEATS is in the range from -7 to 10 μatm. These comparison results indicate the reliability of our reconstruction method and output. All the data for this paper are openly and freely available at PANGAEA under the link https://doi.pangaea.de/10.1594/PANGAEA.921210 (Wang et al., 2020).


## 1 Introduction

The ocean plays an important role in absorbing atmospheric $CO_2$ and consequently helps slow down the Earth's global warming (Le Quere et al., 2018a). Over the last half-century the ocean has taken up approximately 24 % of the total emitted $CO_2$ at an increasing rate from $1.0\pm0.5$ Gt C $yr^{-1}$ in the 1960s to $2.6\pm0.6$ Gt C $yr^{-1}$ in 2019 (Friedlingstein et al., 2020; Le Quere

et al., 2018b). The ocean has been found to be responsible for up to 40 % of the decadal variability of $CO_2$ accumulation in the atmosphere (DeVries et al., 2019). However, the regional and global patterns of the oceanic carbon sink vary both spatially and temporally (Doney et al., 2009; Fay and McKinley, 2013; Landschutzer et al., 2014; Le Quere et al., 2010; Rodenbeck et al., 2015; Turi et al., 2014). Thus, it is necessary to improve the spatial-temporal coverage and accuracy of the data in the evaluation of oceanic carbon uptake capacity in order to better understand the global carbon cycle and to better project the

future climate.

The sea–air $CO_2$ flux is primarily determined by the difference in the atmospheric and sea surface partial pressure of $CO_2$ ($pCO_2$). The measurement values of sea surface fugacity of $CO_2$ ($fCO_2$, which is equal to $pCO_2$ corrected for the non-ideal behavior of the gas (Pfeil et al., 2013)) have increased to 28.2 million and are presently available in almost all ocean basins based on the Surface Ocean $CO_2$ Atlas Version 2020 (Bakker et al., 2020). However, for a given year the observations of sea

surface $pCO_2$ may still have sparse spatial coverage. Thus, interpolation and/or extrapolation methods are needed to obtain a complete $pCO_2$ field in space and time over the concerned oceanic areas. Various methods have been applied for this purpose in the past two decades, including statistical interpolation (Chou et al., 2005) and empirical formulas between $pCO_2$ and proxies such as sea surface temperature, salinity, chlorophyll $a$, sea surface height, and mixed layer depth (Boutin et al., 1999; Denvil-Sommer et al., 2019; Jo et al., 2012; Laruelle et al., 2017; Lefevre and Taylor, 2002; Ono et al., 2004; Zhai et al., 2005a).

These studies usually present their $pCO_2$ fields in a monthly time scale and at a $1^{\circ} \times 1^{\circ}$ or even coarser grid. In marginal seas a finer grid resolution is needed to discern influences posed by local forces such as plumes and upwelling.

The South China Sea (SCS) is the largest marginal sea in the western Pacific. Measurements of sea surface $pCO_2$ in the SCS have started as early as 2000 (Zhai et al., 2005b). Seasonal and spatial variations are present in different domains of the SCS (Li et al., 2020; Zhai et al., 2013). However, the data coverage is still so sparse each year that on global compilation maps the

SCS is mostly blank (Bakker et al., 2016; Fay and McKinley, 2013; Takahashi et al., 2009). For example, the summer observations of 2017 cover 7 % of the SCS, and those of 2001 cover only 1 %. Consequently, the observational data themselves cannot quantitatively depict the $pCO_2$ field over the entire SCS basin. Thus, it is necessary to reconstruct a space-time complete $pCO_2$ field in the SCS in order to better assess the $CO_2$ source and sink features in the SCS and to supplement the global $pCO_2$ map.

The purpose of this paper is to demonstrate the feasibility of reconstructing the $pCO_2$ field over the SCS basin from the sparse in situ observations in the SCS with a grid resolution of 0.5°×0.5°, using a method illustrated in the flowchart of Fig. 1. This paper focuses on the $pCO_2$ reconstruction for the summer season. As indicated in Fig. 1, we need to use an auxiliary dataset, the remote-sensing derived $pCO_2$ estimates, e.g., from Bai et al. (2015), to calculate empirical orthogonal functions (EOFs) for spatial patterns of $pCO_2$. The remote sensing $pCO_2$ estimates are relatively complete in the space-time grid but less accurate,

compared with in situ observations. The singular value decomposition (SVD) method is applied to the remote sensing estimates to compute the EOFs. These EOFs form an orthogonal basis for the spectral optimal gridding (SOG) method (Shen et al., 2014, 2017; Gao et al., 2015; Lammlein and Shen, 2018). The method uses a multilinear regression to blend the in situ data (treated as the data of the response variable in the regression) and the EOFs (treated as the explanatory variables) together to reconstruct the complete summer $pCO_2$ field at 0.5°×0.5° over the SCS.

Section 2 will describe the datasets and methods, Section 3 includes results and discussion, and the conclusions are in Section 4.

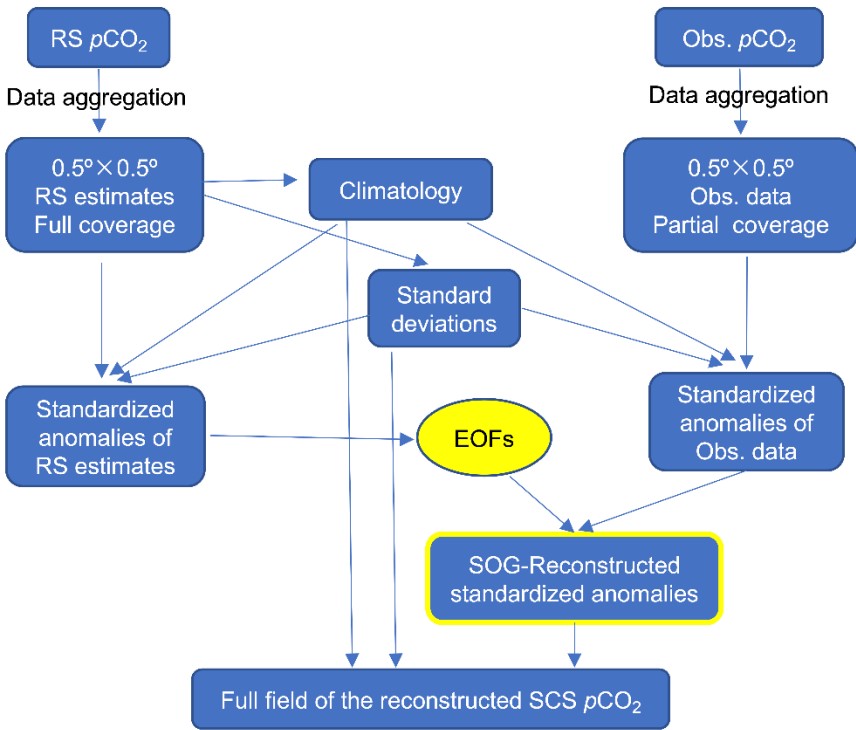

**Figure 1: Reconstruction procedure of the sea surface $pCO_2$ in the SCS. Here, RS $pCO_2$ means the original remote-sensing derived $pCO_2$, Obs. $pCO_2$ represents the original observed in situ $pCO_2$, RS estimates are the grid-aggregated remote-sensing derived $pCO_2$,**

**and Obs. data are the grid-aggregated observed $pCO_2$. The standard deviations are the temporal standard deviation of the RS $pCO_2$ estimates on each grid box.**

## 2 Datasets and methods

### 2.1 Observed data in the SCS

In the SCS, the underway sea surface $pCO_2$ data are hardly available for every month of each year, so we decided to compile
the data seasonally. This study focuses on the summer data since the greatest temporal coverage of the sampling occurs in
summer. The available underway summer $pCO_2$ data from 2000 to 2017 are compiled in this study and shown in Table 1, in
which the $pCO_2$ data in August 2004, July 2005 and June 2006 are new, obtained continuously with a non-dispersive infrared
gas analyzer (Li-Cor 7000). The summer data are the June-August mean for each year in this period excluding 2002, 2003,
2010, 2011 and 2013 (Li et al., 2020; Zhai et al., 2005a). Thus, we have observed underway $pCO_2$ data for 13 summers during
2000–2017. Figure 2 shows that these data are distributed mainly on the northern shelf and slope, and in the northern-and-
mid basin of the SCS with a different frequency of summer observations on different grid boxes. The observational data were
aggregated onto 0.5º×0.5º grid boxes in the (5–25º N, 109–122º E) region of the SCS. The aggregation used a simple space-
time average of the data in a grid box. The aggregated data for the 13 summers are shown in Fig. 3, which shows the distribution
pattern of the observed underway $pCO_2$ data of each year. The aggregated $pCO_2$ in general falls in the range of 160–480 µatm
with relatively larger spatial variation nearshore and smaller spatial variability in the basin. In addition, the large differences
are apparent in the spatial coverage from year to year. For example, in the summer of 2007 the observed underway $pCO_2$ data
cover a spatial range of 12º in latitude and 13º in longitude, 231 grid boxes with data that cover 22 % of SCS. The data fall in
the range of 281–480 µatm. In the summer of 2017 the observed data cover a spatial range of 13º in latitude and 6º in longitude,
77 grid boxes with data that cover 7 % of SCS. The data are in the range of 279–440 µatm. The summer of 2000 has only 5
grid boxes (covering 0.5 % SCS) with data in the range of 400–425 µatm. The lowest observational $pCO_2$ values appear on
the northern SCS shelf due to the influence of the Pearl River plume (See Fig. 2), where nutrient-stimulated phytoplankton
uptake consumes $CO_2$. The relatively high sea surface $pCO_2$ values occur mainly in the basin, which are often higher than the
atmospheric $pCO_2$ (Li et al., 2020; Zhai et al., 2013). The high $pCO_2$ values off the northeastern coast of SCS and the southern
coast of Hainan Island in the summer of 2007 are consistent with local upwelling occurrences, which bring $CO_2$-enriched water
from the subsurface (Li et al., 2020). In the summer of 2012, the spatial coverage is 7º in latitude and 9.5º in longitude. The
$pCO_2$ data are in the range of 191–480 µatm with the lowest value appearing on the northwestern shelf of SCS due to the
Jianjiang River plume and the highest values occurring on the northeast shelf and off the eastern coast of the Hainan Island
due to upwelling (Gan et al., 2015; Jing et al., 2015). Some other data, for example, in the summer of 2000, however, are
relatively localized so that no certain spatial pattern is shown before the reconstruction. Our reconstruction results will help
display the spatial patterns of the complete sea surface $pCO_2$ field.

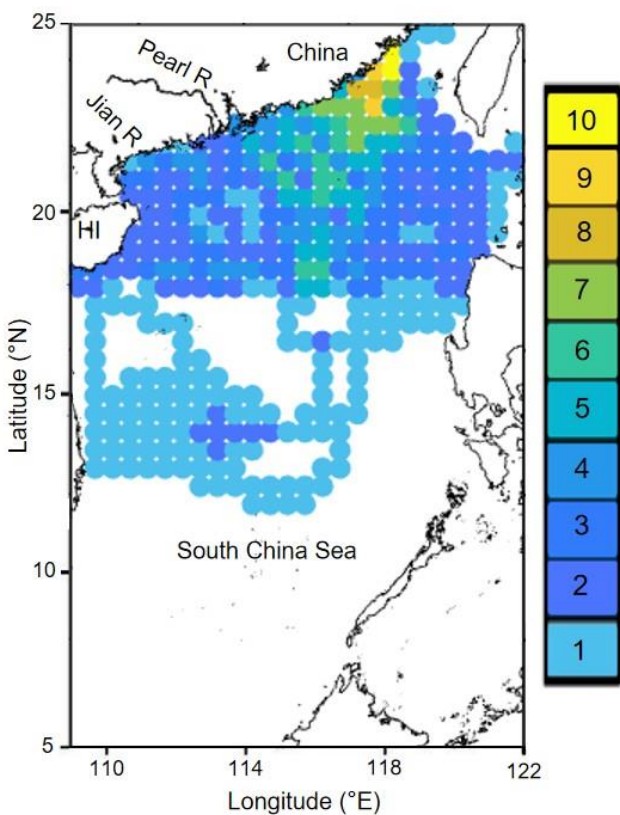

**Figure 2: The number of summers with underway sea surface $p$CO$_2$ observations in the SCS in years 2000–2017. HI represents Hainan Island, Jian. R. is the Jianjiang River, and Pearl R. represents the Pearl River.**

**Table 1. Underway sea surface $p$CO$_2$ data in summer in the SCS compiled in this study.**

| Year | Cruise time | Data source |
|------|-------------|-------------|
| 2000 | July 2000 | Zhai et al., 2005a |
| 2001 | June 2001 | Zhai et al., 2005a |
| 2004 | July–Aug. 2004 | Zhai et al., 2013; This study |
| 2005 | July 2005 | This study |
| 2006 | June 2006 | This study |
| 2007 | July–Aug. 2007 | Zhai et al., 2013 |
| 2008 | July–Aug. 2008 | Li et al., 2020 |
| 2009 | Aug. 2009 | Li et al., 2020 |
| 2012 | July–Aug. 2012 | Li et al., 2020 |
| 2014 | June 2014 | Li et al., 2020 |
| 2015 | July–Aug. 2015 | Li et al., 2020 |
| 2016 | June 2016 | Li et al., 2020 |
| 2017 | June 2017 | Li et al., 2020 |


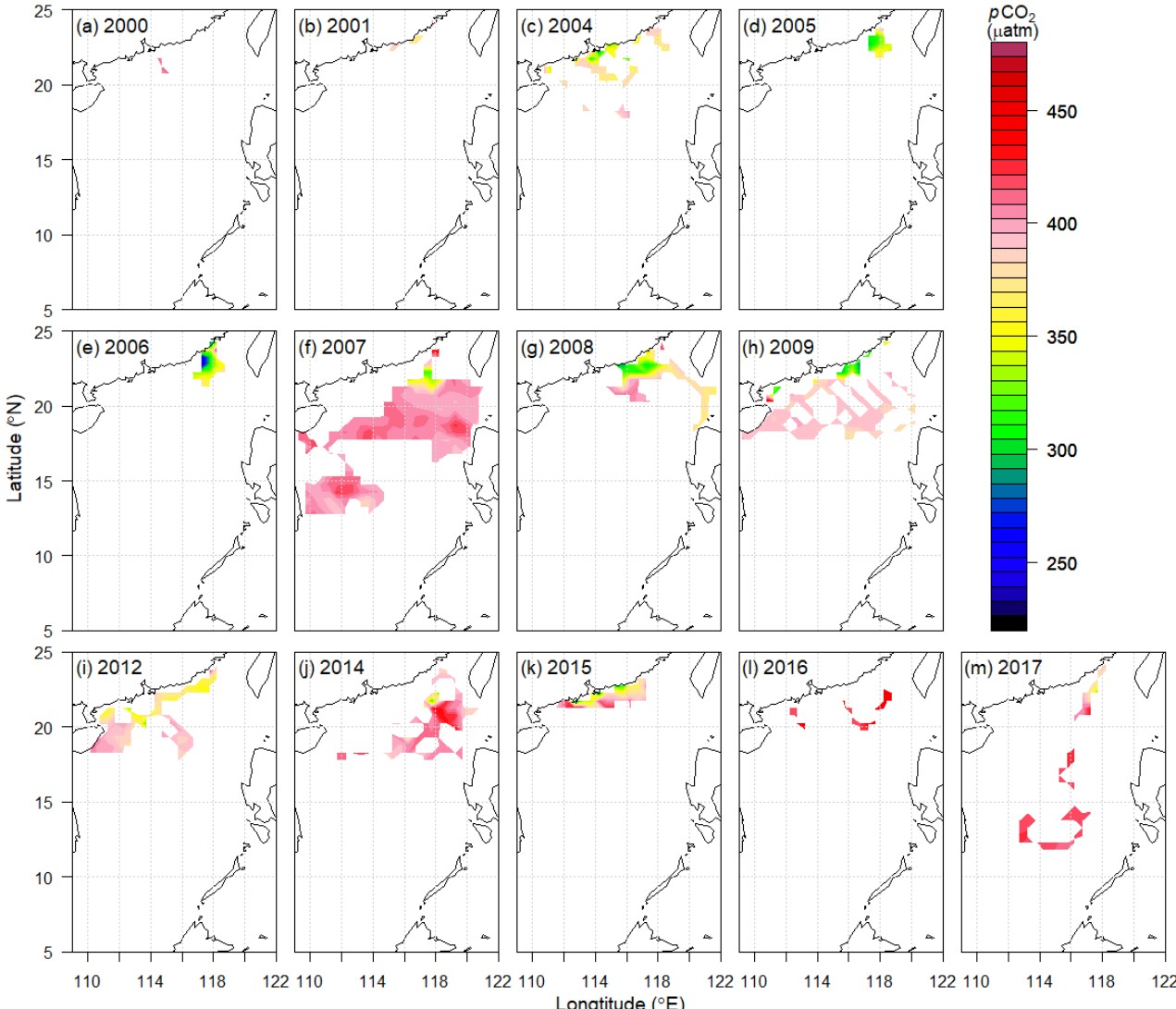

**Figure 3: The aggregated in situ observational $p\mathrm{CO_2}$ data in 0.5º×0.5º grid boxes in the SCS in the 13 summers during years 2000–2017.**

As a dataset for our reconstruction validation, we calculated the sea surface $p\mathrm{CO_2}$ from the observed temperature, salinity,
total alkalinity, dissolved inorganic carbon, phosphate, and silicate at Station SEATS (18º N, 116º E) in the northern basin of
the SCS in the summer of 2007, 2009, 2012, 2014, and 2017. The nutrient sample collection and measurement are described
in Du et al. (2013, 2017). The samples of total alkalinity and dissolved inorganic carbon were collected and measured following
the same procedure in Guo et al. (2015). The calculation of $p\mathrm{CO_2}$ was made using the program of Lewis and Wallace (1998),
in which the apparent dissociation constants for carbonic acid of Mehrbach et al. (1973) refit by Dickson and Millero (1987)
and the dissociation constant for bisulfate ion from Dickson (1990) were employed. Another sea surface $p\mathrm{CO_2}$ dataset

calculated in the same way at Station SEATS in the summer of 2000, 2001, 2004, and 2006 was compiled from Lui et al. (2020).

## 2.2 Remote-sensing derived sea surface $p$CO$_2$ data

The satellite remote-sensing derived sea surface $p$CO$_2$ in the SCS were estimated for the years of 2000–2014 using a "mechanic
semi-analytical algorithm" (MeSAA) developed by Bai et al. (2015a). In the summer of SCS, the thermodynamic, mixing and biological effects on the sea surface $p$CO$_2$ were parameterized in the MeSAA algorithm as a function of major controlling factors derived from multiple satellite-derived sea surface temperature, colored dissolved organic matter, and chlorophyll. The spatial resolution of the remote-sensing derived $p$CO$_2$ estimates is $1´×1´$. These estimates were aggregated into $0.5°×0.5°$ grid boxes in our study region (5–25º N, 109–122º E). As shown in Fig. 4, the gridded remote-sensing derived $p$CO$_2$ data cover
almost all the areas of the SCS (See the boxes of RS $p$CO$_2$ and RS estimates full coverage in Fig. 1). We made a validation study for the remote-sensing derived $p$CO$_2$ by comparison with the observed underway $p$CO$_2$ (Fig. 5). In general, most of the remote-sensing derived $p$CO$_2$ overestimate the sea surface $p$CO$_2$, but not more than 50 µatm. The root-mean-square-error (RMSE) falls in the range from 12.8–89.0 µatm with a median of 33.8 µatm (Table 2). The RMSE values are high in the years when the underway data covered only the shelf regions. With the MeSAA algorithm, the derived $p$CO$_2$ dataset represents the
major CO$_2$ variation in large scales. However, variations shown by these remote-sensing derived $p$CO$_2$ are much less than those shown by the observed $p$CO$_2$ data because the current MeSSA algorithm does not consider some local processes, such as eddies and potentially different carbonate system patterns in coastal areas. Larger spatial variations are expected especially in areas influenced by river plumes. This makes it necessary to reconstruct a $p$CO$_2$ field not only from the remote-sensing derived $p$CO$_2$, but also constrained by the observed in situ $p$CO$_2$ data from cruises.

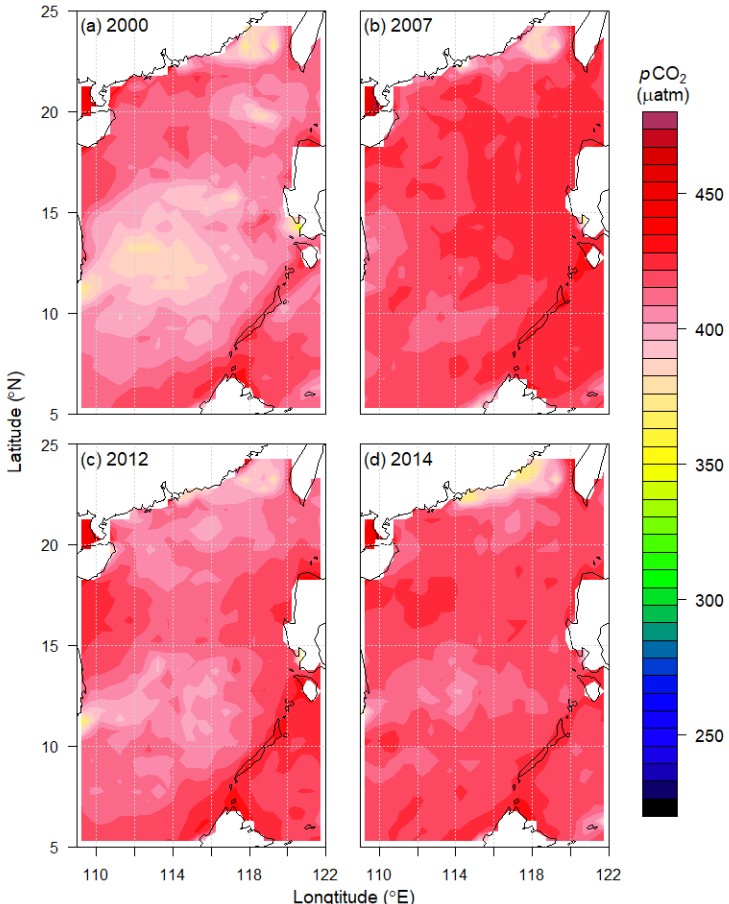


**Figure 4: Remote-sensing derived sea surface $p\mathrm{CO_2}$ in summer in selected years from 2000 to 2014.**

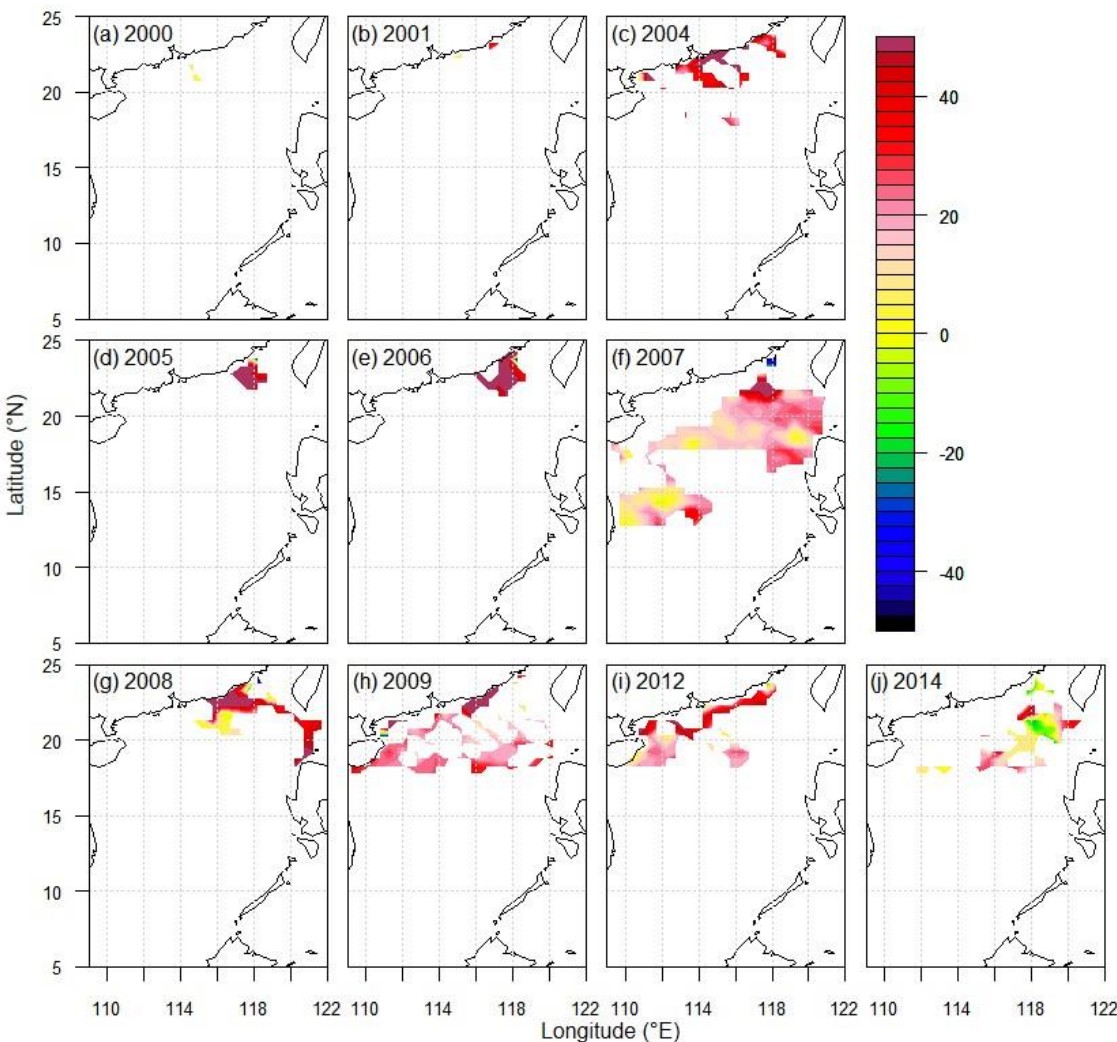

**Figure 5: The difference between the remote-sensing derived summer $p$CO$_2$ estimates and the observed underway $p$CO$_2$ (unit: µatm) in 2000, 2001, 2004–2009, 2012, and 2014.**

## 2.3 Reconstruction method

Figure 1 is a flowchart of our method. We used the remote-sensing derived data to compute the EOFs for the SOG reconstruction. The grid with 0.5°×0.5° resolution covered from 5° to 25° N and from 109° to 122° E with 1040 grid boxes in total. The land area data were marked with NaN. The data were arranged in a 1040×15 space-time matrix with rows for grid boxes and columns for time. Then, we removed the 143 land grid boxes from the data, and computed the climatology and standard deviation for the remaining 897 non-NaN grid boxes from the 15 years of remote-sensing derived data from 2000 to 2014. The standardized anomalies were computed for each grid box using the remote-sensing derived data minus the climatology and subsequently dividing the difference by the standard deviation. The singular value decomposition (SVD)

method was applied to the standardized anomalies in the space–time matrix to compute the EOFs. The results are shown in Section 3. The climatology and standard deviation calculated from the remote-sensing derived data were also used to compute the standardized anomalies of the observed data, which were used as the response variable in the SOG regression reconstruction. Following the reconstruction of the standardized anomalies, the remote-sensing derived climatology and standard deviation were then used to produce the full field as the final reconstruction result.

The SOG reconstruction method is basically a multilinear regression model for the space-time field at grid box $x$ and time $t$, expressed as follows:

$$P(x,t) = \beta_0(t) + \sum_{m \in \mathcal{M}} \beta_m(t) E_m(x) / \sqrt{a(x)} + e(x,t), \tag{1}$$

Here, $P(x,t)$ is the response variable whose data are the standardized anomalies of the observed data, $\beta_0(t)$ is the regression intercept, $\beta_m(t)$ is the regression coefficient for the $m$th EOF $E_m(x)$, the least square estimator of $\beta_m(t)$ is denoted by $b_m(t)$, $a(x) = \cos(\phi_x)$ is the area-factor, $\phi_x$ is the centroid's latitude, expressed in radian, of the grid box $x$, and $e(x,t)$ is the regression error. The error is assumed to be normally distributed with zero mean and has an independent error variance

$$\varepsilon^2(x,t) = E[e^2(x,t)], \tag{2}$$

where $E$ denotes the mathematical operation of expected value. The explanatory variables in the above multilinear regression are $E_m(x)$, computed from the area-weighted standardized anomalies of the remote-sensing derived data. The anomalies were written as an 897×15 space–time data matrix. The SVD method was applied to this matrix to compute the spatial patterns, which are EOFs, the temporal patterns, which are principal components (PCs), and their corresponding variances. $\mathcal{M}$ is the set of EOFs selected for our regression reconstruction. It contained eight EOFs for every year except 2000, which had only four EOFs because the year had only five grid boxes with observed underway data.

For a given year, the grid boxes with observed data are known. Then, the linear regression model can be computed based on the observed data $P(x_d, t)$ and the EOFs in the grid boxes $x_d$ with the observed data $E_m(x_d)$. For example, the year 2002 had only 17 grid boxes with the observational data: $x_1, x_2, \ldots, x_{17}$. The data in these 17 boxes were used to estimate the intercept $\beta_0(t)$ and coefficients $\beta_m(t)$ of the regression. With the estimates $b_0(t)$ and $b_m(t)$, $m \in \mathcal{M}$, the reconstructed standardized anomalies are expressed as

$$\hat{P}(x,t) = b_0(t) + \sum_{m \in \mathcal{M}} b_m(t) E_m(x) / \sqrt{a(x)}, \tag{3}$$

where $x$ runs through the entire 893 grid boxes over our study region in the SCS. These anomalies were converted to the full field by adding the climatology and multiplying the standard deviation computed from the remote-sensing derived data for each of the 893 grid boxes. In this way, the full reconstructed field was produced and is presented in Section 3.

Many computer software packages are available to compute the EOFs using SVD and to compute multilinear regressions. This paper chose to use R, a computer program language that has become a popular data science tool in the last few years for this purpose. The R computer codes and their required files for this paper are freely available at https://github.com/Hqin2019/pCO2-reconstruction

SOG usually uses the first few EOFs, or the first *M* EOFs that account for more than 80 % of the total variance, or determined by response data via a correlation test (Smith et al., 1998). Here, *M* is the size of set $\mathcal{M}$. The current paper used eight EOFs that explain 87 % of the total variances (Fig. 6).

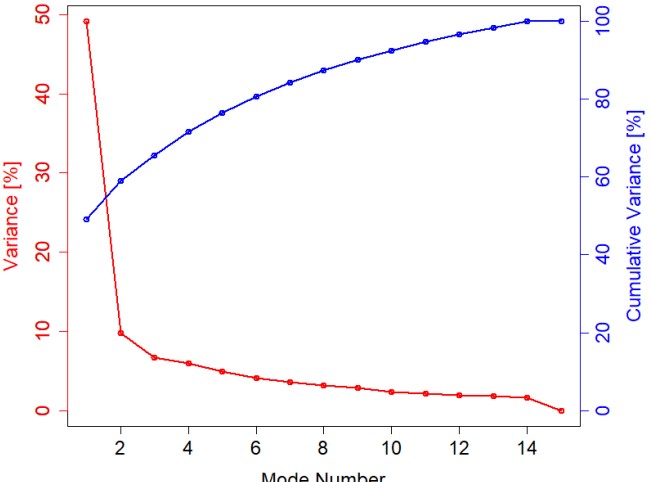

Figure 6: The percentage variances and cumulative variances based on the summer remote-sensing derived $pCO_2$ data for the period
of years 2000–2014.

## 3 Results and discussion

### 3.1 EOFs and PCs

EOF1 demonstrates the mode of an average level of $pCO_2$ with lower or higher values near the coastal regions of China mainland (Fig. 7a). This mode accounts for 49 % of the variance, which indicates the dominance of the average field and hence
a small overall spatial variation, except in the coastal regions. The remote-sensing derived $pCO_2$ data support this mode well. EOF2 shows a north-south dipole (Fig. 7b), which is supported by the observed data shown in Fig. 3, particularly in the summer of 2017, showing lower values in the north on the shelf and slope and higher values in the south in the ocean basin. The minimum values in the north occur where the Pearl River plume dominates (Li et al., 2020; Zhai et al., 2013). EOF3 shows an east–west pattern (Fig. 7c), in addition to the north-south dipole in EOF2. EOF3 thus reflects a spatial variation of a smaller
scale. This pattern is consistent with that influenced by coastal upwelling along the northeast China coast and off eastern Hainan Island (Gan et al., 2015; Jing et al., 2015).

The PCs are temporal stamp of the occurrence of the spatial patterns. PC1 basically shows the temporal trend (Fig. 8d). It has been concluded that surface SCS $pCO_2$ has an increasing trend with time (Tseng et al., 2007). PC2 indicates the strength of the north-south dipole. This strength seems to be related to the strength and extent of the Pearl River plume on the northern
shelf and slope (Bai et al., 2015b; Li et al., 2020; Zhai et al., 2013). PC3 shows the temporal variation corresponding to the east-west spatial pattern of EOF3.

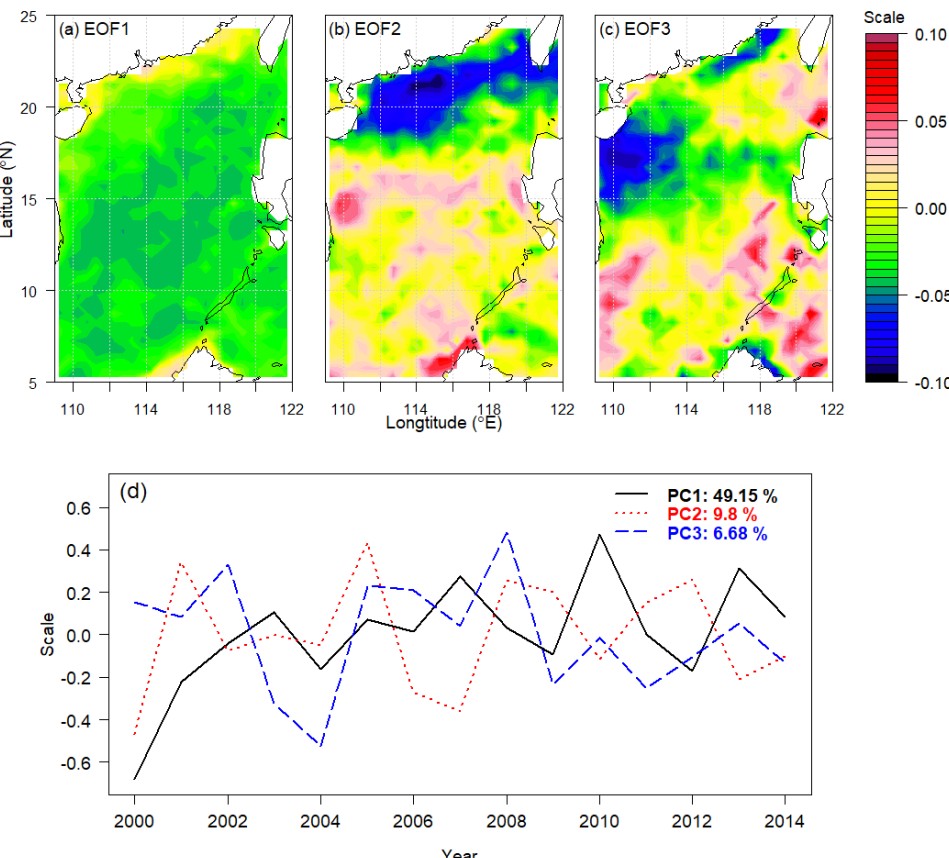

**Figure 7: EOFs and PCs of the remote-sensing derived $p$CO$_2$ estimates. (a)–(c) EOFs, and (d) PCs.**

## 3.2 Reconstruction results in the SCS

Figure 8 shows that the reconstructed $p$CO$_2$ fields in the SCS have successfully displayed the spatial patterns of the observed $p$CO$_2$ and in general are consistent with previous studies (Li et al., 2020; Zhai et al., 2013). Relatively low values appear in the northern coastal region where the Pearl River plume is dominant in summer and generally high values occur in the mid and southern basins.

The reconstructions have taken the advantages of both the observed underway data for retaining spatial and temporal variations 220 and the remote-sensing derived data for EOF patterns. By default, the reconstructed field has fidelity to the in situ data, because the SOG reconstruction method is a fit of EOFs to the in situ data. The reconstruction is, thus, consistent with the in situ observations. When the in situ data cover a sufficiently large area and hence provide a proper constraint to the EOF fitting through the SOG procedure, the reconstruction result is more faithful to the reality. For example, the reconstructions of the summers of 2004, 2007, 2009, 2012, 2014–2017 nicely demonstrate the spatial $p$CO$_2$ patterns (Figs. 8c, f, h, i–m) that are 225 consistent with observations (Li et al., 2020; Zhai et al., 2013) and ocean dynamics (Gan et al., 2015; Jing et al., 2015).

When the observational data are scarce, as long as the in situ data provide a proper constraint to the EOFs, the reconstruction can still yield reasonable results. For example, the summer of 2001 has few in situ data, but its reconstruction, with an RMSE of 7.3 µatm between the reconstructed data and the observed data, appears reasonable (Fig. 8b).

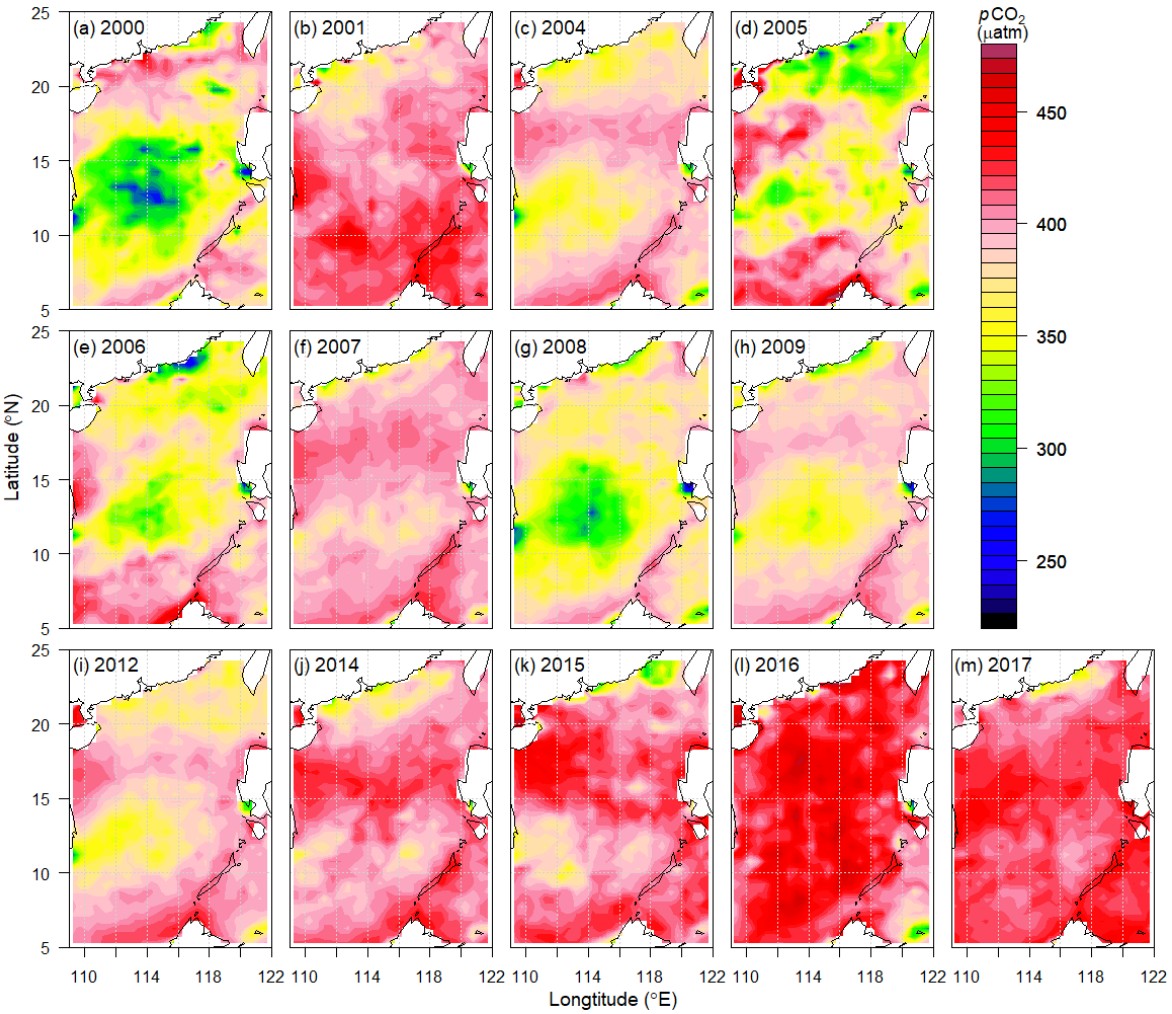

**Figure 8: Reconstructed summer $p$CO$_2$ fields for the years 2000–2017 in the SCS.**

In cases of extreme data scarcity, the reconstruction may not be reliable. For example, the reconstructed data in the summer of 2000 appear to be of poor quality (Fig. 8a) since the relatively low values in the mid SCS basin may not be realistic. These poorly reconstructed data may be due to the poor spatial coverage of the in situ $p$CO$_2$ data in the summer of 2000, which had only 5 grid boxes with data (Fig. 3a). These 5 boxes are all located together and cover only 0.5 % of the SCS. Similarly, the reconstructed $p$CO$_2$ data for the summers of 2005, 2006, and 2008 are not well constrained by the in situ $p$CO$_2$ data that cover only the northern shelf and slope of the SCS so that the reconstructed $p$CO$_2$ in the mid basin are less than 350 µatm (Figs. 8d, e, g). These small values are unlikely since the sea surface $p$CO$_2$ in the basin is generally higher than the atmospheric $p$CO$_2$

(380–420 μatm) (Li et al., 2020; Zhai et al., 2013). Another cause of the less ideal reconstruction results for the summers of 2005, 2006, and 2008 may be the large spatial variations in the in situ data. These variations, such as those for the summer of 2008 (Fig. 3g), in the in situ data can cause a large deviation of the regression coefficients because the linear regression is not robust.

The reconstruction results have demonstrated the feasibility of the SOG reconstruction of the sea surface $p$CO$_2$ over the SCS, as long as the in situ data provide a proper constraint to the EOFs. The percentage of the in situ data coverage needs not necessarily be large. However, large spatial variations in the situ data can distort the reconstruction and lower the quality of reconstruction, because the linear regression method is not robust.

## 3.3 Reconstruction validation and uncertainty quantification

To quantitatively validate our reconstruction, we conducted a leave-one-out cross-validation study: Withholding a grid box datum, making the reconstruction using the remaining in situ data, and computing the difference between the withheld datum and the reconstructed datum at the same grid box. This was repeated for every grid box with in situ data for each year. The final cross-validation result is output as RMSE (Table 2). The maximum RMSE is 5.2 μatm, which occurred in 2006 when there were only 25 grid boxes with in situ $p$CO$_2$ data and the in situ data had the largest spatial standard deviation, 49.4 μatm, among the 13 years under consideration. The minimum RMSE is 2.4 μatm, which occurred in 2017 with 77 in situ data grid boxes and a spatial standard deviation of 17.6 μatm for the in situ data. This accuracy is very good compared to the spatial standard deviation of the in situ data in the same year. Compared to the 2006 data, a more accurate reconstruction for 2017 is expected because of more grid boxes with in situ data and smaller spatial variability. This is supported by the cross-validation result. The spatial standard deviation of the reconstructed data is in the range of 2.1–6.6 μatm. The cross-validation RMSEs are in the range of 2.4–5.2 μatm. We thus conclude that the reliability of our reconstruction is well supported by the leave-one-out cross-validation result.

We have also considered other types of cross-validations, such as leaving out data in half of the study region. A numerical test was made for the following situation: Leaving out the western or eastern half of the data in a year, making the reconstruction using the remaining half of data, and computing the RMSE between the removed data and the reconstructed data. The analysis was done for the years with better spatial coverage: 2007, 2009, and 2012. When the western halves (longitude < 115.5º E) of data in these years were removed, the resulted RMSEs were 2.77, 4.46, and 3.82 μatm for 2007, 2009, and 2012, respectively. When the eastern halves (longitude >115.5º E) of data were removed, the RMSEs were 4.32, 3.66, and 3.55 μatm in 2007, 2009, and 2012, respectively. These RMSEs fall within the range of the RMSEs of the leave-one-out cross validation. These numerical results are another confirmation of the reliability of our reconstruction.

**Table 2: The RMSE between the remote-sensing derived $p$CO$_2$ estimates and the observed underway $p$CO$_2$ data (RMSE$_{RS}$), of the cross-validation (RMSE$_{CV}$), and between the reconstructed $p$CO$_2$ and the observed underway $p$CO$_2$ data (RMSE$_{RC}$) (unit: µatm).**

| Year | 2000 | 2001 | 2004 | 2005 | 2006 | 2007 | 2008 | 2009 | 2012 | 2014 | 2015 | 2016 | 2017 |
|------|------|------|------|------|------|------|------|------|------|------|------|------|------|
| RMSE$_{RS}$ | 12.8 | 20.2 | 47.9 | 65.7 | 89.0 | 25.1 | 43. 8 | 36.8 | 30.7 | 24.2 | NaN | NaN | NaN |
| RMSE$_{CV}$ | NaN | 2.8 | 3.1 | 4.9 | 5.2 | 2.9 | 4.2 | 2.9 | 2.8 | 4.2 | 4.3 | 3.2 | 2.4 |
| RMSE$_{RC}$ | 0.01 | 7.3 | 19.7 | 16.3 | 31.7 | 16.5 | 26.1 | 20.4 | 15.5 | 18.8 | 27.8 | 13.0 | 12.8 |

The uncertainty in our reconstruction was quantified by grid-by-grid comparisons of the reconstructed $p$CO$_2$ with the observed $p$CO$_2$ in two ways. One is the comparison with the observed underway data (Fig. 9). The difference between the reconstructed data and the observed underway data mostly falls within the range from -30 to 30 µatm (Fig. 9). The greatest deviation from the underway data appears near the coast, likely due to the lack of some typical patterns in coastal areas transferred via EOFs from the remote-sensing estimates. The RMSE between the reconstructed data and the observed underway data is no larger than 31.7 µatm with a median of 16.5 µatm, which is smaller than the RMSE between the remote-sensing derived $p$CO$_2$ and the underway data with the relative difference between the two RMSEs (Rows 1 and 3 in Table 2) at least 29 %. When comparing the $p$CO$_2$ data produced by Jo et al. (2012) in the northern SCS by a neural network approach in the summer of years 2004–2007 with the underway $p$CO$_2$, the resultant RMSE falls in the range from 32.6–44.5 µatm and is twice as much as the median RMSE between our reconstructed $p$CO$_2$ and the underway $p$CO$_2$ (Table 2). Another uncertainty quantification for our reconstruction is to compare with the $p$CO$_2$ calculated from long-term observations at Station SEATS (Fig. 10). The difference between the reconstructed $p$CO$_2$ and the observed data at Station SEATS ranges from -7 to 10 µatm with the relative difference from -1.5 to 2.1 %. Both comparisons confirm that our reconstruction results are reliable.

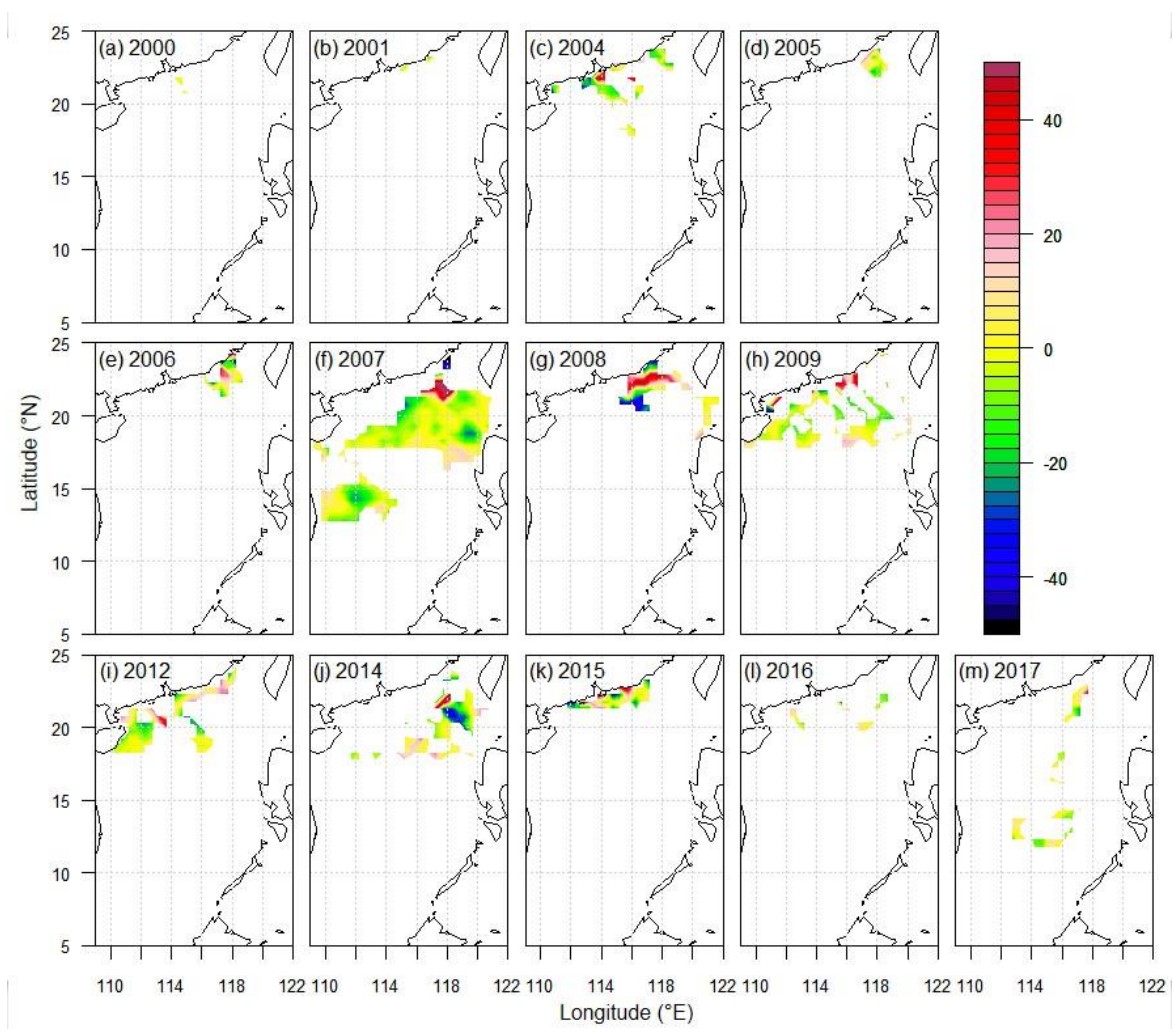

**Figure 9: The difference between the reconstructed summer $p$CO$_2$ and the observed underway $p$CO$_2$ (unit: μatm) in 2000, 2001, 2004–2009, 2012, and 2014–2017.**

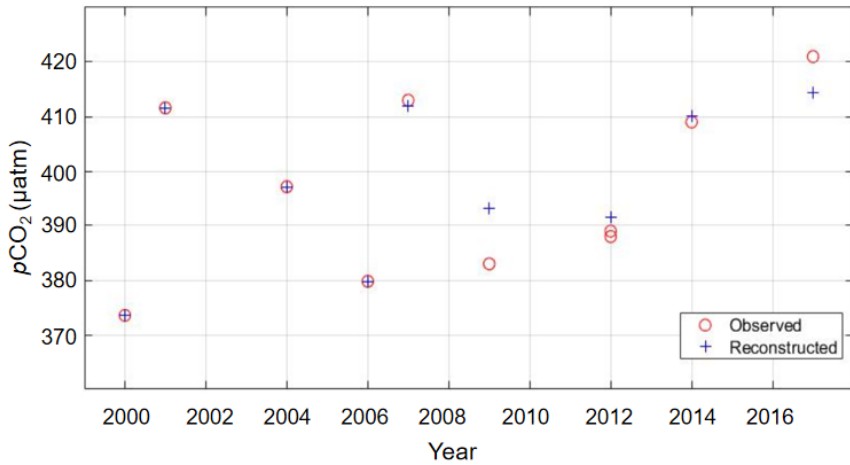

**Figure 10: The comparison between the summer sea surface pCO₂ calculated from the observations and those from our reconstruction at Station SEATS (18º N, 116º E). The *p*CO₂ data calculated from the observations in years of 2000, 2001, 2004, and 2006 are from Hui et al. (2020).**

As an application of our reconstruction and a validation, we examine the temporal trend of sea surface $p$CO₂ over the SCS. The rate based on the linear temporal trend of the spatial average of the reconstructed sea surface $p$CO₂ over the SCS from 2000 to 2017 is 2.4±0.8 µatm per year (See Fig. 11a). It is lower than the rate of $f$CO₂ increase (4 µatm per year) for the period of 1999–2003 (Tseng et al., 2007), while higher than the rate of $p$CO₂ increase for the period of 1998–2006 (0.8 µatm per year) at Station SEATS (Lui et al., 2020). The differences between their rates and ours exist because (a) our rate is a spatial average

in summer and their rates are based on data collected in spring, summer, fall, and winter at a basin station, and (b) the period to derive our rate is much longer than theirs. Using the summer data in Lui et al. (2020), the rate we estimated from the year of 2000, which is the beginning year of our data, to the year of 2006 at Station SEATS is 2.5±1.0 µatm per year. Although the period of 2000–2006 is much shorter than our period of 2000–2017, the summer rate at Station SEATS is almost the same as our rate based on the reconstructed data over the SCS. When compared with the summer rate of observed $p$CO₂ at the Hawaii

Ocean Time-Series Station (Station HOT) (22º45´ N, 158º W) in the North Pacific, which is 2.0±0.2 µatm per year over 2000–2017 (Dore et al., 2009) (See Fig. 11b), our rate is consistent with the rate in the North Pacific. The consistency of the trend in our reconstructed sea surface $p$CO₂ over the SCS with the local trend at Station SEATS and the North Pacific trend at Station HOT confirms that our reconstruction is reasonable.

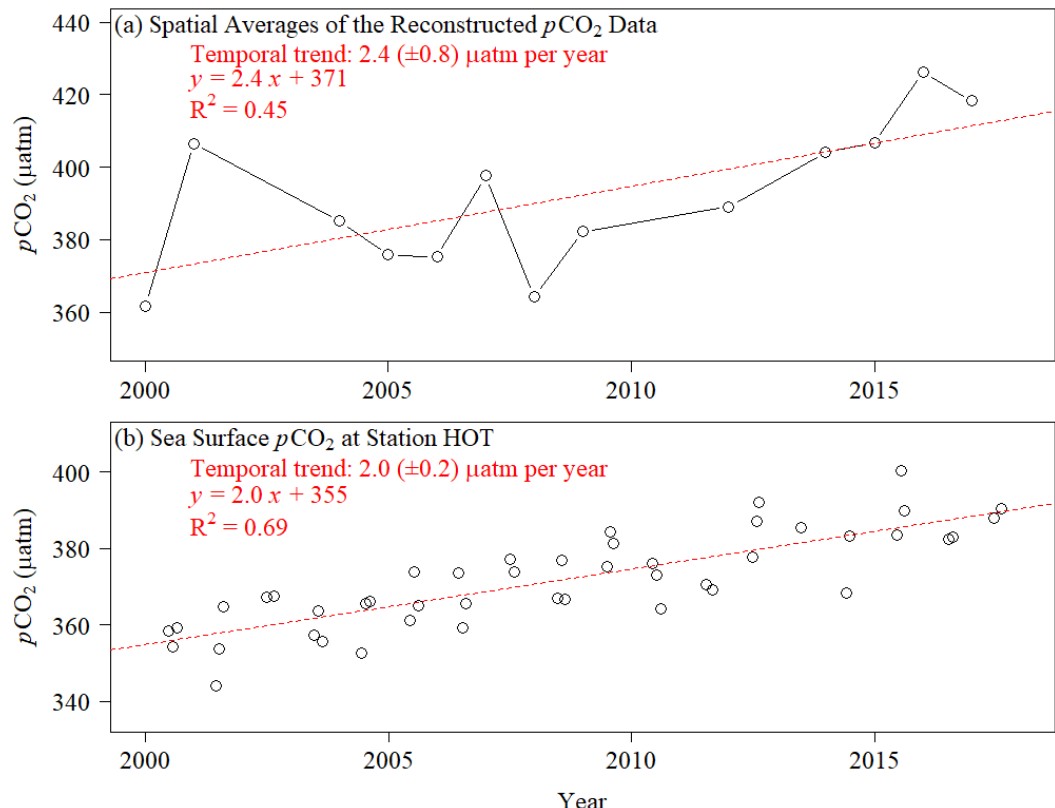

**Figure 11: (a) Time series and linear trend of the spatial averages of the reconstructed summer $pCO_2$ data in the SCS in the period of years 2000–2017; (b) Summer sea surface $pCO_2$ at Station HOT in years 2000–2017 adapted from Dore et al. (2009).**

### 3.4 Outliers of the observed data in the reconstruction

The SOG method is basically a linear regression method, which is known to be sensitive to the outliers of the response data. Some outliers, whether due to observational biases or extreme events, can cause a large change in the regression coefficients, and hence the regression results, and can even make the regression results outside the physically valid domain, such as negative $pCO_2$ values in the reconstructed data. Although we cannot conclude that the outliers of $3\sigma$ away from the mean in the observed data are due to data biases, we have decided not to use them in our reconstruction to avoid the unphysical reconstruction results. Table 3 shows the 14 outlier entries excluded from our response data for regression. These outliers are located in the region of (21.25–23.25° N, 113.25–116.75° E). This region is near the Pearl River Estuary. Thus, these extremely low $pCO_2$ values may result from the Pearl River plume where the observed $pCO_2$ can be very low. These very low values, such as at least $3\sigma$ away from the mean, may cause a very large gradient in the observed $pCO_2$. Our reconstruction has excluded these extremely low values influenced by the river plumes. Our reconstructed data may therefore overestimate the $pCO_2$ values in the Pearl River Estuary and its nearby region.

**Table 3. Outliers excluded from the SOG reconstruction.**

| Year | Grid ID | Latitude (N) | Longitude (E) | $pCO_2$ (µatm) |
|------|---------|--------------|---------------|----------------|
| 2006 | 926 | 22.75° | 116.75° | 208 |
| 2006 | 952 | 23.25° | 116.75° | 197 |
| 2009 | 896 | 22.25° | 114.75° | 212 |
| 2009 | 923 | 22.75° | 115.25° | 217 |
| 2012 | 836 | 21.25° | 110.75° | 248 |
| 2014 | 873 | 21.75° | 116.25° | 191 |
| 2014 | 874 | 21.75° | 116.75° | 219 |
| 2016 | 841 | 21.25° | 113.25° | 265 |
| 2016 | 842 | 21.25° | 113.75° | 272 |
| 2016 | 868 | 21.75° | 113.75° | 239 |
| 2016 | 869 | 21.75° | 114.25° | 205 |
| 2016 | 870 | 21.75° | 114.75° | 216 |
| 2016 | 896 | 22.25° | 114.75° | 210 |
| 2016 | 897 | 22.25° | 115.25° | 274 |

## 4 Data availability

The gridded underway sea surface $pCO_2$ data, the remote-sensing derived sea surface $pCO_2$ estimates, and the reconstruction result data are openly and freely available at PANGAEA under the link https://doi.pangaea.de/10.1594/PANGAEA.921210 (Wang et al., 2020).

## 5 Conclusions

This study has demonstrated the feasibility of using the SOG method to reconstruct the sea surface $pCO_2$ data into regular grid boxes. We compiled the observed underway and remote-sensing derived sea surface $pCO_2$ data in the SCS in summer over the period of 2000–2017 and aggregated these data with a grid resolution of 0.5°×0.5° for reconstruction. The SOG method based on the multilinear regression was applied to reconstruct the space-time complete $pCO_2$ field in the SCS. The method took the EOFs calculated from the remote-sensing derived $pCO_2$ as the explanatory variables and treated the observed $pCO_2$ as the response variable. The EOFs reflect reasonably well the general spatial pattern of the sea surface $pCO_2$ in the SCS and reveal features affected by regional physical forcing such as the river plume and coastal upwelling in the northern SCS. As long as the in situ data provide a proper constraint to the EOFs, the reconstructed $pCO_2$ fields are, in general, consistent with the patterns of the observed $pCO_2$ and demonstrate relatively low values along the north coast affected by the Pearl River plume and consistently high values in the ocean basin of the SCS. The leave-one-out cross-validation result validates our reconstruction with an RMSE smaller than the spatial standard deviation of the observed underway data in the same year. The grid-by-grid comparison of the reconstructed summer $pCO_2$ with the observed underway $pCO_2$ has an RMSE smaller than that of the remote-sensing derived $pCO_2$, as well as that of the neural network produced $pCO_2$ in the same year. Moreover, our

reconstructed $pCO_2$ compares well with the $pCO_2$ calculated from observations around Station SEATS in the northern basin

of the SCS. These comparisons confirm that our reconstruction is reliable. The temporal rate of our reconstructed sea surface $pCO_2$ over the SCS is consistent with the local rate at Station SEATS and the North Pacific rate at Station HOT, which further validates our reconstruction. These reconstructed $pCO_2$ fields provide full spatial coverage of the sea surface $pCO_2$ of the SCS in summer over a temporal scale of almost two decades and therefore help fill the long-lasting blanks on the global sea surface $pCO_2$ map. Thus, the reconstruction products will help improve the accuracy of the estimate of the oceanic $CO_2$ flux of the

largest marginal sea of the western Pacific so as to better constrain the global oceanic carbon uptake capacity.

Although the SOG method can optimize the information from both the in situ data and the remote-sensing derived data, the reliability of the reconstructed results is still limited by the observed data. When the observed data are limited to only a few grid boxes in a small region, the reconstruction results may not be realistic. Additional constraints have to be considered.

**Author contribution**

Minhan Dai conceptualized and directed the field program of the in situ observations. Baoshan Chen and Xianghui Guo participated in the in situ data collection. Yan Bai provided the remote-sensing derived data. Guizhi Wang, Yao Chen and Samuel S. P. Shen developed the reconstruction method, wrote the Matlab and R codes, analyzed the data, and plotted the figures. Zhixuan Wang participated in the uncertainty analysis of the reconstruction and plotted some of the figures. Huan Qin developed the data repository, and revised and tested the R codes. Guizhi Wang and Samuel S. P. Shen wrote the manuscript.

All the authors contributed to the original writing, editing and revisions of the manuscript.

**Competing interests**

The authors declare that they have no conflict of interest.

**Acknowledgements**

We thank Hon-Kit Lui for the communication about the data at Station SEATS. We are thankful to Weidong Zhai for providing

the underway $pCO_2$ data in his publications and to Young-Heon Jo for providing the neural-network produced $pCO_2$ estimates in his published paper. We appreciate the constructive comments from the two anonymous reviewers that have helped improve this paper. Aiqin Han, Tao Huang, and Lifang Wang helped in nutrient sample collection and measurements and Liguo Guo, Wenping Jing, Yi Xu, Wei Yang, and Nan Zheng helped in the sample collection and measurements of total alkalinity and dissolved inorganic carbon at Station SEATS. The work by Guizhi Wang, Yan Bai, Xianghui Guo, and Minhan Dai was

supported by grants from the Ministry of Science and Technology of China (2015CB954001, 2009CB421200). The observed underway $pCO_2$ data in 2004, 2005 and 2006 were collected under the support of the National Natural Science Foundation of

China (40521003). Acknowledgement is for the data support from "National Earth System Science Data Sharing Infrastructure, National Science & Technology Infrastructure of China (http://www.geodata.cn)".

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
