# Peer review of "Feasibility of reconstructing the summer basin-scale sea surface partial pressure of carbon dioxide from sparse in situ observations over the South China Sea"

_Earth System Science Data, 2020_

## Referee Comment (RC1) · Anonymous Referee #1 · 25 Sep 2020

The subject is important but the manuscript suffers from two major flaws. Fortunately, both could be amended.

My major concern is the reliability of the result. The manuscript gave a rate of pCO2 increase of 2.38 uatm/yr, which is very high. Unfortunately, no uncertainty was given. Judged by the large scatter of the data(Fig. 8a) the standard deviation of the rate must be very large. Note other studies, for instance, that of Lui et al. (2020, Transient carbonate chemistry in the expanded Kuroshio region, in Changing Asia Pacific Marginal Seas, pp 307-320) gave a much lower increasing rate of only 0.8 uatm/yr at

the SEATS station. I fully recognize that different sampling locations, sampling periods, and sampling frequency could contribute to large differences in the results. Yet, exactly because of this the result must be qualified and compared with other studies. In addition, there ought to be other independent checks of the pCO2 data generated by the satellite chlorophyll data. There is an abundance of alkalinity, DIC, and pH data in various parts of the South China Sea, especially at SEATS. It would be relatively easy to generate pCO2 from these data to check model-derived pCO2.

My second major concern is the coverage of the data. The manuscript covers data from only 13 years, and in most years the region covered was very small. In fact, none of the cruise tracks covers the southern South China Sea. It seems that the authors used only their own data but why not include other people's data as well? For instance, the open-access SOCAT database covers tracks in the southern South China Sea.

One minor issue is that the title does not reflect correctly that only summer data were covered.

---

## Referee Comment (RC2) · Anonymous Referee #2 · 7 Nov 2020

The authors have used a remote-sensing-based pCO2 field to derive EOFs, fit those EOFs to in situ pCO2 observations collected over almost 2 decades, and then used the scaled EOFs to estimate the full surface pCO2 record in the South China Sea.

It is a very interesting paper and many parts of it are clearly communicated, but it is also incomplete. The method validation and uncertainty quantification are missing. These should be an entire section of the paper and not just an added sentence or two, so the paper should be returned to the authors for major revisions.

It is a bit unclear whether this paper is presenting new data along with a new method or just a new method. There are two cruises in 2005 and 2006 with a reference given as "this paper" and, if these data sets are truly being published for the first time in this paper, then the manuscript should highlight that there are new data in the abstract. This would raise the value of this paper if there are indeed new data being made available along with the analysis. I might have missed the text that explained this.

The part of the paper that deals with the $pCO_2$ mapping approach is not yet complete because the authors have not assessed the uncertainties of their approach. I recommend one or two exercises. First, the approach should be repeated after removing some of the in situ $pCO_2$ measurements. Each cruise should be removed, one at a time. After removing a cruise, the analysis should be conducted using only the remaining data. Then the withheld cruise can be used to quantify how good of a job the mapping procedure does at reconstructing the withheld cruise. This should be repeated for every cruise in the dataset to get bulk statistics. If there is only one cruise worth of data in each year, then (I believe this reconstruction wouldn't work and instead) large swaths of latitude/longitude should be removed from the cruises and the remaining data should be used to reconstruct the missing data. This will allow the errors in the approach to be quantified. Second, if a model is available for the South China Sea that has $pCO_2$, then the model can also have the Bai et al. 2015 approach applied, be subsampled where the cruise measurements are, and be analyzed in the same way proposed here. This will reveal both the point-by-point reconstruction errors and allow the uncertainties for the overall $pCO_2$ average estimate, for example, to be quantified. Currently, the validation is left as an unsupported statement that the results look about right, which is insufficient for publication of a paper describing a quantitative method.

There are other smaller problems that should also be addressed if a revised version of the paper is submitted:

1. The model should not be used in any region where there is no fitting data. This

includes most of the South China Sea south of ~12.5 N.

2. There should be an assessment of how good of a job the Bai et al. approach does at reproducing the in situ observations in a RMSE and bias sense. The estimates from this approach should be compared to the measurements from the data sets that are used here (and that Bai et al. did not use to design their routine). If the Bai et al. approach gives a different average pCO2 than the in situ measurements, then the climatology created from the remote sensing product should not be used to generate the Standardized Anomalies of Obs. Data (as indicated in Figure 1). I believe an independent climatology would then be needed. Otherwise, a significant average bias would have to be compensated by a large average value for one or more EOFs. In a best-case scenario, that would be EOF 1, but if, for example, the observations were mostly found in the dark blue patch of Figure 6c then the resulting reconstruction would be problematic. It seems likely that a large average value of EOF 3, which is highly variable spatially, would then be fit to the measurements to fix a homogenous bias between the in situ and remote-sensing records. This is just one example of the kinds of problems that could occur if the Bai reconstruction doesn't adequately resolve the mean or the variability. If nothing else, the Bai et al. validation should be discussed in this paper.

It would also be interesting to see how this approach compares to competing approaches, for example a neural network that relates the in situ pCO2 measurements to seawater property values that can be measured using remote sensing. This approach is more commonly used in global reconstructions. The Bai et al. approach is another clear competing approach.

Specific comments:

15: consider deleting "capacity"

23: "The reconstructions always agree with observations." Delete or quantify this statement. The agreement is not absolute.

[Figure]

28: The ocean

36: The sea-air CO2 flux is the negative of the ocean carbon uptake, so this sentence is partially tautological.

37: This sentence has several language errors. It also needs to be better-quantified or referenced. What is the decorrelation length scale for pCO2 generally? How much of the ocean is constrained by those measurements alone without the newly proposed spatio-temporal mapping techniques? Mostly, I think a reference should be added to this sentence that supports this statement.

57: References needed for RS pCO2 here.

Figure 1. What is meant by "standard deviations"? Standard deviations of grid values, or deviations of values within each grid cell?

Figure 2. Consider changing this map to a 2 dimensional histogram showing number summers with measurements (probably with colored bins).

109: These estimates were... change "data" to "estimates" in this section since pCO2 is not measured. In figure 1 as well.

138: Where is this symbol used?

141: how many EOFs were used? Say here.

182: This only shows the fields. One must compare this field to other figures to get an indication of how well the reconstruction performs. A plot showing differences between observed and reconstructed values is required.

187: It is not enough to say "we fit the model to the data, so it fits the data." Statistics of goodness-of-fit should be presented. Furthermore, demonstrating that the method works requires withholding several cruises worth of data from the training data set and then using those cruises to verify that the method reconstructs the withheld data. Statistics and plots are required to quantify how well the reconstruction does.

195: What is meant by reasonable?

205: it is unclear what is meant by "the large spatial gradient of in situ data."

214: 2.383 is given to excessive precision. An attempt should be made to quantify the uncertainty and the data should be reported to the appropriate precision.

229: why would a higher rate be expected in a marginal sea? I would argue that 2.4 uatm/year is completely within expectations of the atmospheric $pCO_2$ trend over this time period given the large uncertainties in this approach and the likely natural variability in surface ocean $pCO_2$.

---

## Author Comment (AC1) · 2 Dec 2020

The subject is important but the manuscript suffers from two major flaws. Fortunately, both could be amended.

My major concern is the reliability of the result. The manuscript gave a rate of pCO2 increase of 2.38 uatm/yr, which is very high. Unfortunately, no uncertainty was given. Judged by the large scatter of the data (Fig. 8a) the standard deviation of the rate must be very large. Note other studies, for instance, that of Lui et al. (2020, Transient carbonate chemistry in the expanded Kuroshio region, in Changing Asia Pacific Marginal Seas, pp 307-320) gave a much lower increasing rate of only 0.8 uatm/yr at the SEATS station. I fully recognize that different sampling locations, sampling periods, and sampling frequency could contribute to large differences in the results. Yet, exactly because of this the result must be qualified and compared with other studies. In addition, there ought to be other independent checks of the pCO2 data generated by the satellite chlorophyll data. There is an abundance of alkalinity, DIC, and pH data in various parts of the South China Sea, especially at SEATS. It would be relatively easy to generate pCO2 from these data to check model-derived pCO2.

Response: Thank you for your comments and input.

First, we have quantified the uncertainty of our rate of increase as 2.4 ± 0.8 $\mu$tam/yr, where 0.8 $\mu$tam/yr is the standard error of the rate. See Figure R1.

[Figure]

Figure R1: (a) Time series and linear trend of the spatial averages of the reconstructed summer $p$CO$_2$ data in the period of 2000–2017; (b) Summer sea surface $p$CO$_2$ at Station HOT in 2000–2017 adapted from Dore et al. (2009).

Second, the rate in Lui et al. (2020) of 0.8 µatm/yr is for the period of 1998-2006 and was calculated based on the data of spring, summer, fall, and winter. The reason for their much slower rate might be due to the peak $pCO_2$ value approximately 405 µatm in 1999, as shown in Fig. 16.7 in their paper. This peak value at an earlier time forced a lower rate. We re-calculated the rate using their summer-only data from the year of 2000, which is the beginning year of our data, to the year of 2006, we obtained a rate of 2.5±1.0 µatm/yr, which is almost the same as our rate of 2.4±0.8 µatm/yr.

Third, regarding the question about independent checks for the remote-sensing derived $pCO_2$ estimates, we have compared our observed underway $pCO_2$ with the remote-sensing derived estimates as a quality check. See Figure R2 and Table R1. In general, most of the remote-sensing derived $pCO_2$ overestimate the sea surface $pCO_2$ by no more than 50 µatm. The root-mean-square-errors (RMSE) between the remote-sensing derived $pCO_2$ and the observed underway $pCO_2$ fall in the range of 12.8-89.0 µatm.

[Figure]

Figure R2: The difference between the remote-sensing derived $pCO_2$ estimates and the observed underway $pCO_2$ data in years 2000, 2001, 2004-2009, 2012 and 2014 (unit: µatm).

Table R1: The RMSE between the remote-sensing derived $pCO_2$ estimates and the observed underway $pCO_2$ data (unit: µatm).

| Year | 2000 | 2001 | 2004 | 2005 | 2006 | 2007 | 2008 | 2009 | 2012 | 2014 |
|------|------|------|------|------|------|------|------|------|------|------|
| RMSE | 12.8 | 20.2 | 47.9 | 65.7 | 89.0 | 25.1 | 43.8 | 36.8 | 30.7 | 24.2 |

Fourth, we have used the $pCO_2$ data calculated from alkalinity and DIC data observed at Station SEATS and nearby stations from Liu et al. (2020) and our own database to check the reconstructed $pCO_2$. See Figure R3. The temporal pattern of the reconstructed data is basically consistent with that of the $pCO_2$ data calculated from the observed alkalinity and DIC data. The difference between the reconstructed data and those calculated from observed data falls within $\pm 10.2$ μatm and the relative error is within $\pm 2.1\%$. This comparison between the reconstructed data and an independent dataset calculated from observed data around Station SEATS indicates that our reconstruction is reliable.

We will include a description of the calculation and the observed data used in the calculation in the revised data section of the paper.

[Figure]

Figure R3: The comparison between the $pCO_2$ calculated from the observed total alkalinity and DIC and those from our reconstruction around Station SEATS (18 °N, 116 °E). (a) Locations of the observation stations and the area where the reconstructed $pCO_2$ was selected for comparison. The red circles are observation stations and the black rectangle indicates the area where the reconstructed data are used in comparison, (b) The comparison between the $pCO_2$ data calculated from the observed DIC and total alkalinity and those from our reconstruction. Red circles represent the reconstructed data and the blue crosses represent the $pCO_2$ calculated from the observed total alkalinity and DIC. The difference is $pCO_{2R}$ - $pCO_{2C}$, where $pCO_{2R}$ is the reconstructed $pCO_2$ and $pCO_{2C}$ is calculated from the observations, and the relative error is $(pCO_{2R}-pCO_{2C})/pCO_{2C}\times100\%$.

My second major concern is the coverage of the data. The manuscript covers data from only 13 years, and in most years the region covered was very small. In fact, none of the cruise tracks covers the southern South China Sea. It seems that the authors used only their own data but why not include other people's data as well? For instance, the open-access SOCAT database covers tracks in the southern South China Sea. One minor issue is that the title does not reflect correctly that only summer data were covered.

Response: The word 'summer' will be added before 'basin-scale' in the title. Our method allows us to include any group's data if they are available.

As for the data temporal coverage, to our knowledge the underway summer $pCO_2$ data are

available by this date only for 13 years. A $p$CO$_2$ dataset calculated from observed total alkalinity and DIC is present for June 2010 in the northern South China sea in Guo et al. (2015). We have considered this dataset. However, we are still examining an outlier of the dataset. When the outlier and consistency issues of this dataset are resolved, we may include their data in our future reconstruction. Of course, we will also include any group's data when they become available. Observed data from other groups are published either at one or two buoys or stations, e.g., Liu et al. (2020) and Yu et al. (2020) or in other seasons, e.g., Xu et al. (2016). However, the data at one or two buoys or stations cannot provide a spatial coverage needed for the reconstruction. Although the SOCAT database has tracks in the southern South China Sea, it does not have summer data for the region. We plan to make reconstruction in other seasons in the future and we will then include these data.

As for the spatial coverage, the poor coverage of observed data in the South China Sea, especially in the southern South China Sea, is exactly the main reason for us to make the reconstruction. The purpose of our work is to reconstruct a complete $p$CO$_2$ field in the South China Sea.

References

Dore, J. E., Lukas, R., Sadler, D. W., Church, M. J., and Karl, D. M.: Physical and biogeochemical modulation of ocean acidification in the central North Pacific, P. Natl. A. Sci. USA, 106, 12235–12240, doi:10.1073/pnas.0906044106, 2009.

Guo, X. H., and Wong, G. T. F.: Carbonate chemistry in the Northern South China Sea Shelf-sea in June 2010, Deep-Sea Res. Part II, 117, 119-130, doi:10.1016/j.dsr2.2015.02.024, 2015.

Lui, H.-K., Chen, C.-T. A., Hou, W.-P., Yu, S., Chan, J.-W., Bai, Y., and He, X.: Transient Carbonate Chemistry in the Expanded Kuroshio Region, In: Chen, C.-T., and Guo, X. (eds) Changing Asia-Pacific Marginal Seas. Atmosphere, Earth, Ocean & Space. Springer, Singapore, doi:10.1007/978-981-15-4886-4_16, 2020.

Xu, X., Yu, P., Cai, X. Pan, J., Hu, J., and Zhang, H.: Distributions of the partial pressure of carbon dioxide and sea-air CO$_2$ flux in the western South China Sea in autumn, J. Tropical Oceanogr., 35(3), 55-64, doi:10.11978/2015035, 2016.

Yu, P., Wang, Z. A., Churchill, J., Zheng, M., Pan, J., Bai, Y., and Liang, C.: Effects of typhoons on surface seawater $p$CO$_2$ and air-sea CO$_2$ fluxes in the northern South China Sea, J. Geophys. Res., 125, doi:10.1029/2020JC016258, 2020.

---

## Author Comment (AC2) · 4 Dec 2020

The authors have used a remote-sensing-based $pCO_2$ field to derive EOFs, fit those EOFs to in situ $pCO_2$ observations collected over almost 2 decades, and then used the scaled EOFs to estimate the full surface $pCO_2$ record in the South China Sea.

It is a very interesting paper and many parts of it are clearly communicated, but it is also incomplete. The method validation and uncertainty quantification are missing. These should be an entire section of the paper and not just an added sentence or two, so the paper should be returned to the authors for major revisions.

Response: Thank you for bringing up the issues of validation and uncertainty quantification. We first addressed the cross validation issue. The regression-based reconstruction is often valid when outliers are not present. Our reconstruction follows this approach. Nonetheless, we agree with you and have conducted a cross-validation check of our reconstruction. The maximum RMSE of our cross-validation is 5.22 µatm, which occurred in 2006 when there were only 25 grid boxes with in situ $pCO_2$ data and which had the largest spatial standard deviation, 49.40 µatm, among the 13 years under consideration. This accuracy is very good compared to the spatial standard deviation of the in situ data in the same year. The temporal standard deviation of the reconstructed data is in the range of 2.12- 6.60 µatm. The cross-validation RMSEs are in the range of 2.43-5.22 µatm. We thus conclude that the reliability of our reconstruction is well supported by the cross-validation result. We will include our cross-validation method and result in the revised paper.

Second, we addressed the uncertainty issue. We made grid-by-grid comparisons between the observed $pCO_2$ and reconstructed $pCO_2$ in two ways. One is comparison with observed underway data (see Figure R1) and the other is comparison with $pCO_2$ calculated from observed DIC and total alkalinity around Station SEATS (18 °N, 116 °E) (see Figure R2). The RMSE between the reconstructed data and the observed underway data is in the range from 0.01-31.67 µatm (see Table R1). The difference between the reconstructed data and the observed data around Station SEATS ranges from -7 to 10 µatm with the relative error within 2.1%. Both comparisons will be provided as other ways of validation in the revised paper.

[Figure]

Figure R1: Difference between the reconstructed $pCO_2$ and the observed underway $pCO_2$ in years of 2000, 2001, 2004-2009, 2012, 2014-2017 (unit: μatm).

[Figure]

Figure R2: The comparison between the $pCO_2$ calculated from the observed total alkalinity and DIC and those from our reconstruction around Station SEATS (18 °N, 116 °E). (a) Locations of the observation stations and the grids on which the reconstructed $pCO_2$ was selected for comparison. The red circles are observation stations and the blue pluses represent the centers of the reconstruction grids, (b) The comparison between the $pCO_2$ data calculated from the observed DIC

and total alkalinity and those from our reconstruction. The red circles represent the $pCO_2$ calculated from the observed total alkalinity and DIC and the blue pluses represent the reconstructed data. The difference is $pCO_{2R}$ - $pCO_{2C}$, where $pCO_{2R}$ is the reconstructed $pCO_2$ and $pCO_{2C}$ is calculated from the observations, and the relative error is $(pCO_{2R}-pCO_{2C})/pCO_{2C} \times 100\%$.

Table R1: The RMSE between the reconstructed and the observed underway $pCO_2$ data ($RMSE_{RC}$) and between the remote-sensing derived $pCO_2$ estimates and the observed underway $pCO_2$ data ($RMSE_{RS}$)(unit: $\mu atm$).

| Year | 2000 | 2001 | 2004 | 2005 | 2006 | 2007 | 2008 | 2009 | 2012 | 2014 | 2015 | 2016 | 2017 |
|---|---|---|---|---|---|---|---|---|---|---|---|---|---|
| $RMSE_{RC}$ | 0.01 | 7.27 | 19.72 | 16.28 | 31.67 | 16.50 | 26.14 | 20.41 | 15.48 | 18.82 | 27.83 | 13.04 | 12.76 |
| $RMSE_{RS}$ | 12.82 | 20.15 | 47.94 | 65.69 | 88.97 | 25.12 | 43.79 | 36.80 | 30.73 | 24.20 | NaN | NaN | NaN |

It is a bit unclear whether this paper is presenting new data along with a new method or just a new method. There are two cruises in 2005 and 2006 with a reference given as "this paper" and, if these data sets are truly being published for the first time in this paper, then the manuscript should highlight that there are new data in the abstract. This would raise the value of this paper if there are indeed new data being made available along with the analysis. I might have missed the text that explained this.

Response: This paper is presenting new data along with a new method. In the revised data the new data in 2005 and 2006 will be highlighted in the abstract and in the main text. In addition, $pCO_2$ data from literature and calculated from our alkalinity and DIC around a basin station, SEATS (18 °N, 116 °E), will be presented to compare with the reconstructed $pCO_2$ as a way of validation in the revised paper.

The part of the paper that deals with the $pCO_2$ mapping approach is not yet complete because the authors have not assessed the uncertainties of their approach. I recommend one or two exercises. First, the approach should be repeated after removing some of the in situ $pCO_2$ measurements. Each cruise should be removed, one at a time. After removing a cruise, the analysis should be conducted using only the remaining data. Then the withheld cruise can be used to quantify how good of a job the mapping procedure does at reconstructing the withheld cruise. This should be repeated for every cruise in the dataset to get bulk statistics. If there is only one cruise worth of data in each year, then (I believe this reconstruction wouldn't work and instead) large swaths of latitude/longitude should be removed from the cruises and the remaining data should be used to reconstruct the missing data. This will allow the errors in the approach to be quantified. Second, if a model is available for the South China Sea that has $pCO_2$, then the model can also have the Bai et al. 2015 approach applied, be subsampled where the cruise measurements are, and be analyzed in the same way proposed here. This will reveal both the point-by-point reconstruction errors and allow the uncertainties for the overall $pCO_2$ average estimate, for example, to be quantified. Currently, the validation is left as an unsupported statement that the results look about right, which is insufficient for publication of a paper describing a quantitative method.

Response: Again, you have suggested a cross-validation procedure. As aforementioned, we have conducted a leave-one-out cross-validation study: Withholding a grid box datum, making the reconstruction using the remaining in situ data, and computing the difference between the withheld datum and the reconstructed datum at the same grid box. This is done for every grid box

with in situ data for each year. The final cross-validation result is output as RMSE. The maximum RMSE is 5.22 μatm, which occurred in 2006, and the minimum is 2.43 μatm, which occurred in 2017. The year 2017 has 77 in situ data grid boxes. The spatial standard deviation of the data in 2017 is 17.55 μatm. Compared to the 2006 data described earlier, a more accurate reconstruction for 2017 is expected because of more grid boxes with in situ data and smaller spatial variability. This is supported by the cross-validation.

Your second suggestion can be mathematically proven, because the cross-validation RMSE of reconstruction from the sub-sample of the Bai et al. (2015) complete data is only the truncation error, which is equal to zero or very close to be zero. The reason is that the EOFs computed from Bai et al. (2015) data form a complete basis for the same data. Thus, the original data field can be exactly represented by a linear span of the EOFs.

There are other smaller problems that should also be addressed if a revised version of the paper is submitted:
1. The model should not be used in any region where there is no fitting data. This includes most of the South China Sea south of ~12.5 N.

Response: This is exactly the point that shows the power of the spectral optimal gridding (SOG) method using EOFs, in contrast to the traditional optimal interpolation method, such as kriging and inverse distance weighting. EOFs are a diagonalized representation of the covariance of climate dynamics, and thus are providing a consistency constraint of the $pCO_2$ field. This allows us to use a small number of grid boxes with in situ data to interpolate and extrapolate to the entire region. Of course, the reconstruction accuracy is better when more observed data are available.

2. There should be an assessment of how good of a job the Bai et al. approach does at reproducing the in situ observations in a RMSE and bias sense. The estimates from this approach should be compared to the measurements from the data sets that are used here (and that Bai et al. did not use to design their routine). If the Bai et al. approach gives a different average $pCO_2$ than the in situ measurements, then the climatology created from the remote sensing product should not be used to generate the Standardized Anomalies of Obs. Data (as indicated in Figure 1). I believe an independent climatology would then be needed. Otherwise, a significant average bias would have to be compensated by a large average value for one or more EOFs. In a best-case scenario, that would be EOF 1, but if, for example, the observations were mostly found in the dark blue patch of Figure 6c then the resulting reconstruction would be problematic. It seems likely that a large average value of EOF 3, which is highly variable spatially, would then be fit to the measurements to fix a homogenous bias between the in situ and remote-sensing records. This is just one example of the kinds of problems that could occur if the Bai reconstruction doesn't adequately resolve the mean or the variability. If nothing else, the Bai et al. validation should be discussed in this paper.
It would also be interesting to see how this approach compares to competing approaches, for example a neural network that relates the in situ $pCO_2$ measurements to seawater property values that can be measured using remote sensing. This approach is more commonly used in global reconstructions. The Bai et al. approach is another clear competing approach.

Response: We did a grid-by-grid assessment of the remote-sensing derived $pCO_2$ using the observed underway $pCO_2$ as shown in Figure R3. In addition, a comparison between the observed

$p$CO$_2$ and reconstructed $p$CO$_2$ is provided in Figure R1. Both comparisons will be provided as validations in the revised paper. The RMSEs of the two comparisons are shown in Table R1 and will be provided in the revised paper. Furthermore, $p$CO$_2$ data from literature and calculated from our alkalinity and DIC around a basin station, SEATS (18 $^\circ$N, 116 $^\circ$E), were compared with the reconstructed $p$CO$_2$ as another way of validation as shown in Figure R2 and will be presented in the revised paper.

[Figure]

Figure R3: The difference between the remote-sensing derived $p$CO$_2$ estimates and the observed underway $p$CO$_2$ data in years 2000, 2001, 2004-2009, 2012 and 2014 (unit: µatm).

With regard to EOFs and independent climatology, the mathematical theory is like Fourier expansion of orthogonal polynomials, which can be sine functions, Legendre polynomials, and any set of eigenfunctions of a self-adjoint operator. Thus, EOFs form a complete basis for a data field although they may be different when using different anomalies computed from different climatologies and standard deviations. With different anomalies, variances may be re-distributed to different EOFs due to the different anomaly calculation methods. EOF rotation may help to reorganize certain variances into some specific EOF modes, and hence to provide an explanation of climate dynamics. However, this EOF rotation is not needed for the purpose of reconstruction as long as our EOFs form a complete basis. This completeness is guaranteed by the SVD algorithm for computing our EOFs here.

As for reconstruction using a neural network approach, the data produced by Jo et al. (2012) show an overall RMSE of 32.59-44.52 in summer $p$CO$_2$ reconstruction as validated using the observed

underway data in the northern South China Sea, which overlaps with the RMSE of our reconstruction.

Specific comments:

15: consider deleting "capacity"

Response: The suggestion will be taken in the revised paper.

23: "The reconstructions always agree with observations." Delete or quantify this statement. The agreement is not absolute.

Response: This statement in the revised paper will be changed to "The RMSE between the reconstructed data and the observed underway data is in the range from 0.01-31.67 $\mu$atm and the difference between the reconstructed data and those calculated from observations around Station SEATS ranges from -7 to 10 $\mu$atm with the relative error within 2.1%, both of which indicate a good agreement of our reconstruction with observations."

28: The ocean

Response: The suggestion will be taken in the revised paper.

36: The sea-air $CO_2$ flux is the negative of the ocean carbon uptake, so this sentence is partially tautological.

Response: In the revised paper "helps quantify the oceanic carbon uptake capacity" will be deleted.

37: This sentence has several language errors. It also needs to be better-quantified or referenced. What is the decorrelation length scale for $pCO_2$ generally? How much of the ocean is constrained by those measurements alone without the newly proposed spatio-temporal mapping techniques? Mostly, I think a reference should be added to this sentence that supports this statement.

Response: The language errors of the sentence will be eliminated in the revised paper. The paper Bakker et al. (2016) will be added to the reference list. This paper is the most recent published compilation of measured $pCO_2$.

57: References needed for RS $pCO_2$ here.

Response: The suggestion is taken. Bai et al. (2015) will be added here in the revised paper.

Figure 1. What is meant by "standard deviations"? Standard deviations of grid values, or deviations of values within each grid cell?

Response: It is the temporal standard deviation of the RS $pCO_2$ values on each grid box.

Figure 2. Consider changing this map to a 2 dimensional histogram showing number summers with measurements (probably with colored bins).

Response: This map will be changed to the following Figure R4 in the revised paper.

[Figure]

Figure R4: The number of summers with underway sea surface $pCO_2$ observations in the SCS in the period of 2000-2017. HI represents Hainan Island, Jian. R. is the Jianjiang River, and Pearl R. represents the Pearl River.

109: These estimates were… change "data" to "estimates" in this section since $pCO_2$ is not measured. In figure 1 as well.
Response: The changes will be made here and in Figure 1 in the revised paper.

138: Where is this symbol used?
Response: The symbol $\langle\cdot\rangle$ is only used once for expected value in Eq. (2) in the paper. In the revised version, we will replace the symbol $\langle\cdot\rangle$ by E[ ], which is more commonly used in statistics and science, while $\langle\cdot\rangle$ is a symbol commonly used in the field of theoretical physics.

141: how many EOFs were used? Say here.
Response: The suggestion will be taken in the revised paper. Eight EOFs were used.

182: This only shows the fields. One must compare this field to other figures to get an indication of how well the reconstruction performs. A plot showing differences between observed and reconstructed values is required.
Response: We have taken your suggestion and produced two figures showing differences between observed and reconstructed values. See Figures R1 and R2.

187: It is not enough to say "we fit the model to the data, so it fits the data." Statistics of goodness-of-fit should be presented. Furthermore, demonstrating that the method works requires

withholding several cruises worth of data from the training data set and then using those cruises to verify that the method reconstructs the withheld data. Statistics and plots are required to quantify how well the reconstruction does.

Response: Again, this is a cross-validation issue discussed earlier. We will include our results of cross-validation and uncertainty quantification in the revised paper as shown in our response to the previous comments.

195: What is meant by reasonable?

Response: In the revised paper the RMSE, 7.27 μatm, will be provided here, which indicates that the reconstruction appears reasonable.

205: it is unclear what is meant by "the large spatial gradient of in situ data."

Response: Here it means the large spatial variation of in situ data. The "gradient" is changed to "variation" in the revision.

214: 2.383 is given to excessive precision. An attempt should be made to quantify the uncertainty and the data should be reported to the appropriate precision.

Response: The suggestion is taken. The rate will be given as 2.4±0.8 μatm/yr in the revised paper.

229: why would a higher rate be expected in a marginal sea? I would argue that 2.4 uatm/year is completely within expectations of the atmospheric $pCO_2$ trend over this time period given the large uncertainties in this approach and the likely natural variability in surface ocean $pCO_2$.

Response: Considering the uncertainty in the rate is 0.8 μatm/yr, we agree with the reviewer that our rate is consistent with the trend shown at Station HOT in the Pacific. In the revised paper, this statement will be included and the sentence about "a higher rate be expected in a marginal sea" will be deleted.

References

Bai, Y., Cai, W.-J., He, X., Zhai, W. D., Pan, D., Dai, M., and Yu, P.: A mechanistic semi-analytical method for remotely sensing sea surface pCO2 in river-dominated coastal oceans: a case study from the East China Sea, J. Geophys. Res. 120, 2331–2349, doi:10.1002/2014JC010632, 2015.

Bakker, D. C. E., Pfeil, B. C., Landa, S., Metzl, N., M. O'Brien, K., Olsen, A., Smith, K., Cosca, C., Harasawa, S., Jones, S. D., Nakaoka, S., Nojiri, Y., Schuster, U., Steinhoff, T., Sweeney, C., Takahashi, T., Tilbrook, B., Wada, C., Wanninkhof, R., Alin, S. R., Balestrini, C. F., Barbero, L., Bates, N. R., Bianchi, A. A., Bonou, F., Boutin, J., Bozec, Y., Burger, E. F., Cai, W. J., Castle, R. D., Chen, L. Q., Chierici, M., Currie, K., Evans, W., Featherstone, C., Feely, R. A., Fransson, A., Goyet, C., Greenwood, N., Gregor, L., Hankin, S., Hardman-Mountford, N. J., Harlay, J., Hauck, J., Hoppema, M., Humphreys, M. P., Hunt, C., Huss, B., Ibanhez, J. S. P., Johannessen, T., Keeling, R., Kitidis, V., Kortzinger, A., Kozyr, A., Krasakopoulou, E., Kuwata, A., Landschutzer, P., Lauvset, S. K., Lefevre, N., Lo Monaco, C., Manke, A., Mathis, J. T., Merlivat, L., Millero, F. J., Monteiro, P. M. S., Munro, D. R., Murata, A., Newberger, T., Omar, A. M., Ono, T., Paterson, K., Pearce, D., Pierrot, D., Robbins, L. L., Saito, S., Salisbury, J., Schlitzer, R., Schneider, B., R. Schweitzer, R., Sieger, R., Skjelvan, I., Sullivan, K. F., Sutherland, S. C., Sutton, A. J., Tadokoro, K., Telszewski, M., Tuma, M., van Heuven, S., Vandemark, D., Ward, B., Watson, A. J., and Xu,

S. Q.: A multi-decade record of high-quality $fCO_2$ data in version 3 of the Surface Ocean $CO_2$ Atlas (SOCAT), Earth System Science Data, 8, 383–413, doi:10.5194/essd-8-383-2016, 2016.

Jo, Y. H., Dai, M. H., Zhai, W. D., Yan, X. H., and Shang, S. L.: On the variations of sea surface $pCO_2$ in the northern South China Sea: A remote sensing based neural network approach, J. Geophys. Res., 117, C08022, doi:10.1029/2011JC007745, 2012.

---

## Author Response (AR1)

Our response is in blue with line numbers listed where revisions are made.

Anonymous Referee #1

The subject is important but the manuscript suffers from two major flaws. Fortunately, both could be amended.

My major concern is the reliability of the result. The manuscript gave a rate of pCO2 increase of 2.38 uatm/yr, which is very high. Unfortunately, no uncertainty was given. Judged by the large scatter of the data (Fig. 8a) the standard deviation of the rate must be very large. Note other studies, for instance, that of Lui et al. (2020, Transient carbonate chemistry in the expanded Kuroshio region, in Changing Asia Pacific Marginal Seas, pp 307-320) gave a much lower increasing rate of only 0.8 uatm/yr at the SEATS station. I fully recognize that different sampling locations, sampling periods, and sampling frequency could contribute to large differences in the results. Yet, exactly because of this the result must be qualified and compared with other studies. In addition, there ought to be other independent checks of the pCO2 data generated by the satellite chlorophyll data. There is an abundance of alkalinity, DIC, and pH data in various parts of the South China Sea, especially at SEATS. It would be relatively easy to generate pCO2 from these data to check model-derived pCO2.

Response: Thank you for your comments and input.

First, we have quantified the uncertainty of our rate of increase as 2.4±0.8 $\mu$tam/yr, where 0.8 $\mu$tam/yr is the standard error of the rate, in the revision (Line 287 and Figure 11).

Second, the rate in Lui et al. (2020) of 0.8 $\mu$atm/yr is for the period of 1998-2006 and was calculated based on the data of spring, summer, fall, and winter. The reason for their much slower rate might be due to the peak $p$CO$_2$ value approximately 405 $\mu$atm in 1999, as shown in Fig. 16.7 in their paper. This peak value at an earlier time forced a lower rate. We re-calculated the rate using their summer-only data from the year of 2000, which is the beginning year of our data, to the year of 2006, we obtained a rate of 2.5±1.0 $\mu$atm/yr, which is almost the same as our rate of 2.4±0.8 $\mu$atm/yr. We have added their rate, as well as the reasons for the differences between their rate and ours, and the recalculated rate in the revision (Line 288-294).

Third, regarding the question about independent checks for the remote-sensing derived $p$CO$_2$ estimates, we have compared our observed underway $p$CO$_2$ with the remote-sensing derived estimates as a quality check. In general, most of the remote-sensing derived $p$CO$_2$ overestimate the sea surface $p$CO$_2$ by no more than 50 $\mu$atm (Figure 5 in the revision). The root-mean-square-errors (RMSE) between the remote-sensing derived $p$CO$_2$ and the observed underway $p$CO$_2$ fall in the range of 12.8-89.0 $\mu$atm with a median of 33.8 $\mu$atm (Table 2 in the revision). The RMSE values are high in the years when the underway data covered only the shelf regions. These texts (Line

135-142), one Figure (Figure 5), and one table (Table 2) are added in the revision.

Fourth, we have used the $p$CO$_2$ data calculated from alkalinity and DIC data observed at Station SEATS from Lui et al. (2020) and our own database to check the reconstructed $p$CO$_2$ (Figure 10 in the revision). The difference between the reconstructed $p$CO$_2$ and the observed data at Station SEATS ranges from -7 to 10 $\mu$atm with the relative difference from -1.5 to 2.1 %. This comparison indicates that our reconstruction is reliable. We have added these texts (Line 273-276) and one figure (Figure 10) in the revision.

We have included a description of the calculation and the observed data used in the calculation in the revised data section of the paper (Line 119-127).

My second major concern is the coverage of the data. The manuscript covers data from only 13 years, and in most years the region covered was very small. In fact, none of the cruise tracks covers the southern South China Sea. It seems that the authors used only their own data but why not include other people's data as well? For instance, the open-access SOCAT database covers tracks in the southern South China Sea. One minor issue is that the title does not reflect correctly that only summer data were covered.

Response: The word 'summer' have been added before 'basin-scale' in the title in the revision. Our method allows us to include any group's data if they are available.

As for the data temporal coverage, to our knowledge the underway summer $p$CO$_2$ data are available by this date only for 13 years. A $p$CO$_2$ dataset calculated from observed total alkalinity and DIC is present for June 2010 in the northern South China sea in Guo et al. (2015). We have considered this dataset. However, we are still examining an outlier of the dataset. When the outlier and consistency issues of this dataset are resolved, we may include their data in our future reconstruction. Of course, we will also include any group's data when they become available. Observed data from other groups are published either at one or two buoys or stations, e.g., Lui et al. (2020) and Yu et al. (2020) or in other seasons, e.g., Xu et al. (2016). However, the data at one or two buoys or stations cannot provide a spatial coverage needed for the reconstruction. Although the SOCAT database has tracks in the southern South China Sea, it does not have summer data for the region. We plan to make reconstruction in other seasons in the future and we will then include these data.

As for the spatial coverage, the poor coverage of observed data in the South China Sea, especially in the southern South China Sea, is exactly the main reason for us to make the reconstruction. The purpose of our work is to reconstruct a complete $p$CO$_2$ field in the South China Sea.

References
Dore, J. E., Lukas, R., Sadler, D. W., Church, M. J., and Karl, D. M.: Physical and biogeochemical modulation of ocean acidification in the central North Pacific, P. Natl. A. Sci. USA, 106, 12235–12240, doi:10.1073/pnas.0906044106, 2009.
Guo, X. H., and Wong, G. T. F.: Carbonate chemistry in the Northern South China Sea Shelf-sea in

June 2010, Deep-Sea Res. Part II, 117, 119-130, doi:10.1016/j.dsr2.2015.02.024, 2015.

Lui, H.-K., Chen, C.-T. A., Hou, W.-P., Yu, S., Chan, J.-W., Bai, Y., and He, X.: Transient Carbonate Chemistry in the Expanded Kuroshio Region, In: Chen, C.-T., and Guo, X. (eds) Changing Asia-Pacific Marginal Seas. Atmosphere, Earth, Ocean & Space. Springer, Singapore, doi:10.1007/978-981-15-4886-4_16, 2020.

Xu, X., Yu, P., Cai, X. Pan, J., Hu, J., and Zhang, H.: Distributions of the partial pressure of carbon dioxide and sea-air $CO_2$ flux in the western South China Sea in autumn, J. Tropical Oceanogr., 35(3), 55-64, doi:10.11978/2015035, 2016.

Yu, P., Wang, Z. A., Churchill, J., Zheng, M., Pan, J., Bai, Y., and Liang, C.: Effects of typhoons on surface seawater $p$CO$_2$ and air-sea $CO_2$ fluxes in the northern South China Sea, J. Geophys. Res., 125, doi:10.1029/2020JC016258, 2020.

Anonymous Referee #2

The authors have used a remote-sensing-based $p$CO$_2$ field to derive EOFs, fit those EOFs to in situ $p$CO$_2$ observations collected over almost 2 decades, and then used the scaled EOFs to estimate the full surface $p$CO$_2$ record in the South China Sea.

It is a very interesting paper and many parts of it are clearly communicated, but it is also incomplete. The method validation and uncertainty quantification are missing. These should be an entire section of the paper and not just an added sentence or two, so the paper should be returned to the authors for major revisions.

Response: Thank you for bringing up the issues of validation and uncertainty quantification. We first addressed the cross validation issue. The regression-based reconstruction is often valid when outliers are not present. Our reconstruction follows this approach. Nonetheless, we agree with you and have conducted a cross-validation check of our reconstruction. The maximum RMSE of our cross-validation is 5.2 μatm, which occurred in 2006 when there were only 25 grid boxes with in situ $p$CO$_2$ data and which had the largest spatial standard deviation, 49.4 μatm, among the 13 years under consideration. This accuracy is very good compared to the spatial standard deviation of the in situ data in the same year. The temporal standard deviation of the reconstructed data is in the range of 2.1- 6.6 μatm. The cross-validation RMSEs are in the range of 2.4-5.2 μatm. We thus conclude that the reliability of our reconstruction is well supported by the cross-validation result. We have included our cross-validation method and result in the revised paper (Line 247-258, Table 2).

Second, we addressed the uncertainty issue. We have made grid-by-grid comparisons between the observed $p$CO$_2$ and reconstructed $p$CO$_2$ in two ways. One is comparison with observed underway data (Figure 9 and Table 2 in the revision) and the other is comparison with $p$CO$_2$ calculated from observed DIC and total alkalinity at Station SEATS (18° N, 116° E) (Figure 10 in the revision). The difference between the reconstructed data and the observed underway data mostly falls within the range from -30 to 30 μatm. The greatest deviation from the underway data appears near the coast, likely due to the lack of some typical patterns in coastal areas transferred via EOFs from the remote-sensing estimates. The RMSE between the reconstructed data and the observed underway data is no larger than 31.7 μatm with a median of 16.5 μatm, which is smaller than the RMSE

between the remote-sensing derived $pCO_2$ and the underway data with the relative difference between the two RMSEs (Rows 1 and 3 in Table 2) at least 29 %. The difference between the reconstructed data and the observed data at Station SEATS ranges from -7 to 10 $\mu$atm with the relative error within 2.1 %. Both comparisons have been provided as other ways of validation in the revised paper (Line 264-276, Figures 9 and 10, Table 2).

It is a bit unclear whether this paper is presenting new data along with a new method or just a new method. There are two cruises in 2005 and 2006 with a reference given as "this paper" and, if these data sets are truly being published for the first time in this paper, then the manuscript should highlight that there are new data in the abstract. This would raise the value of this paper if there are indeed new data being made available along with the analysis. I might have missed the text that explained this.

Response: This paper is presenting new data along with a new method. In the revised data the new data in 2004, 2005 and 2006 have been highlighted in the abstract (Line 18-19) and in the main text (Line 86-88). In addition, $pCO_2$ data from literature and calculated from our alkalinity and DIC at Station SEATS (18° N, 116° E) have been presented to compare with the reconstructed $pCO_2$ as a way of validation in the revised paper (Line 119-127 for the data description and Line 273-276 for comparison with the reconstructed estimates).

The part of the paper that deals with the $pCO_2$ mapping approach is not yet complete because the authors have not assessed the uncertainties of their approach. I recommend one or two exercises. First, the approach should be repeated after removing some of the in situ $pCO_2$ measurements. Each cruise should be removed, one at a time. After removing a cruise, the analysis should be conducted using only the remaining data. Then the withheld cruise can be used to quantify how good of a job the mapping procedure does at reconstructing the withheld cruise. This should be repeated for every cruise in the dataset to get bulk statistics. If there is only one cruise worth of data in each year, then (I believe this reconstruction wouldn't work and instead) large swaths of latitude/longitude should be removed from the cruises and the remaining data should be used to reconstruct the missing data. This will allow the errors in the approach to be quantified. Second, if a model is available for the South China Sea that has $pCO_2$, then the model can also have the Bai et al. 2015 approach applied, be subsampled where the cruise measurements are, and be analyzed in the same way proposed here. This will reveal both the point-by-point reconstruction errors and allow the uncertainties for the overall $pCO_2$ average estimate, for example, to be quantified. Currently, the validation is left as an unsupported statement that the results look about right, which is insufficient for publication of a paper describing a quantitative method.

Response: Again, you have suggested a cross-validation procedure. As aforementioned, we have conducted a leave-one-out cross-validation study: Withholding a grid box datum, making the reconstruction using the remaining in situ data, and computing the difference between the withheld datum and the reconstructed datum at the same grid box. This is done for every grid box with in situ data for each year. The final cross-validation result is output as RMSE. The maximum RMSE is 5.2 $\mu$atm, which occurred in 2006, and the minimum is 2.4 $\mu$atm, which occurred in 2017. The year 2017 has 77 in situ data grid boxes. The spatial standard deviation of the data in 2017 is 17.6 $\mu$atm. Compared to the 2006 data described earlier, a more accurate reconstruction for 2017 is expected because of more grid boxes with in situ data and smaller spatial variability. This is

supported by the cross-validation. We have included our cross-validation method and result in the revised paper (Line 247-258, Table 2).

Your second suggestion can be mathematically proven, because the cross-validation RMSE of reconstruction from the sub-sample of the Bai et al. (2015) complete data is only the truncation error, which is equal to zero or very close to be zero. The reason is that the EOFs computed from Bai et al. (2015) data form a complete basis for the same data. Thus, the original data field can be exactly represented by a linear span of the EOFs.

There are other smaller problems that should also be addressed if a revised version of the paper is submitted:
1. The model should not be used in any region where there is no fitting data. This includes most of the South China Sea south of ~12.5 N.
Response: This is exactly the point that shows the power of the spectral optimal gridding (SOG) method using EOFs, in contrast to the traditional optimal interpolation method, such as kriging and inverse distance weighting. EOFs are a diagonalized representation of the covariance of climate dynamics, and thus are providing a consistency constraint of the $pCO_2$ field. This allows us to use a small number of grid boxes with in situ data to interpolate and extrapolate to the entire region. Of course, the reconstruction accuracy is better when more observed data are available.

2. There should be an assessment of how good of a job the Bai et al. approach does at reproducing the in situ observations in a RMSE and bias sense. The estimates from this approach should be compared to the measurements from the data sets that are used here (and that Bai et al. did not use to design their routine). If the Bai et al. approach gives a different average $pCO_2$ than the in situ measurements, then the climatology created from the remote sensing product should not be used to generate the Standardized Anomalies of Obs. Data (as indicated in Figure 1). I believe an independent climatology would then be needed. Otherwise, a significant average bias would have to be compensated by a large average value for one or more EOFs. In a best-case scenario, that would be EOF 1, but if, for example, the observations were mostly found in the dark blue patch of Figure 6c then the resulting reconstruction would be problematic. It seems likely that a large average value of EOF 3, which is highly variable spatially, would then be fit to the measurements to fix a homogenous bias between the in situ and remote-sensing records. This is just one example of the kinds of problems that could occur if the Bai reconstruction doesn't adequately resolve the mean or the variability. If nothing else, the Bai et al. validation should be discussed in this paper.
It would also be interesting to see how this approach compares to competing approaches, for example a neural network that relates the in situ $pCO_2$ measurements to seawater property values that can be measured using remote sensing. This approach is more commonly used in global reconstructions. The Bai et al. approach is another clear competing approach.
Response: We have made a grid-by-grid assessment of the remote-sensing derived $pCO_2$ using the observed underway $pCO_2$ in the revision. In general, most of the remote-sensing derived $pCO_2$ overestimate the sea surface $pCO_2$ by no more than 50 $\mu$atm (Figure 5 in the revision). The root-mean-square-errors (RMSE) between the remote-sensing derived $pCO_2$ and the observed underway $pCO_2$ fall in the range of 12.8-89.0 $\mu$atm with a median of 33.8 $\mu$atm (Table 2 in the revision). The RMSE values are high in the years when the underway data covered only the shelf

regions. These texts (Line 135-142), one Figure (Figure 5), and one table (Table 2) are added in the revision. In addition, a comparison between the observed underway $p$CO$_2$ and reconstructed $p$CO$_2$ has been provided in the revision (Line 264-273, Figure 9). Both comparisons have been provided as validations in the revised paper. The RMSEs of the two comparisons have been shown in Table 2 in the revision. Furthermore, $p$CO$_2$ data from literature and calculated from our alkalinity and DIC at Station SEATS (18° N, 116° E) were compared with the reconstructed $p$CO$_2$ as another way of validation in the revised paper (Line 119-127 for the data description and Line 273-276 for comparison with the reconstructed estimates).

With regard to EOFs and independent climatology, the mathematical theory is like Fourier expansion of orthogonal polynomials, which can be sine functions, Legendre polynomials, and any set of eigenfunctions of a self-adjoint operator. Thus, EOFs form a complete basis for a data field although they may be different when using different anomalies computed from different climatologies and standard deviations. With different anomalies, variances may be re-distributed to different EOFs due to the different anomaly calculation methods. EOF rotation may help to reorganize certain variances into some specific EOF modes, and hence to provide an explanation of climate dynamics. However, this EOF rotation is not needed for the purpose of reconstruction as long as our EOFs form a complete basis. This completeness is guaranteed by the SVD algorithm for computing our EOFs here.

As for reconstruction using a neural network approach, the data produced by Jo et al. (2012) show an overall RMSE of 32.6-44.5 in summer of years 2004-2007, which is twice as much as the median RMSE between our reconstructed $p$CO$_2$ and the underway $p$CO$_2$. We have added this comparison in the revision (Line 270-273).

Specific comments:
15: consider deleting "capacity"
Response: The suggestion has been taken in the revised paper.

23: "The reconstructions always agree with observations." Delete or quantify this statement. The agreement is not absolute.
Response: This statement has been changed in the revised paper to "The RMSE between the reconstructed summer $p$CO$_2$ and the observed underway $p$CO$_2$ is no larger than 31.7 $\mu$atm, in contrast to (a) the RMSE from 12.8–89.0 $\mu$atm between the remote-sensing derived $p$CO$_2$ and the underway data, and (b) the RMSE from 32.6–44.5 $\mu$atm between the neural network produced $p$CO$_2$ and the underway data. The difference between the reconstructed $p$CO$_2$ and those calculated from observations at Station SEATS is in the range from -7 to 10 $\mu$atm. These comparison results indicate the reliability of our reconstruction method and output." (Line 29-34).

28: The ocean
Response: The suggestion has been taken in the revised paper (Line 37).

36: The sea-air CO$_2$ flux is the negative of the ocean carbon uptake, so this sentence is partially tautological.

Response: In the revised paper "helps quantify the oceanic carbon uptake capacity" has been deleted.

37: This sentence has several language errors. It also needs to be better-quantified or referenced. What is the decorrelation length scale for $pCO_2$ generally? How much of the ocean is constrained by those measurements alone without the newly proposed spatio-temporal mapping techniques? Mostly, I think a reference should be added to this sentence that supports this statement.

Response: The language errors of the sentence have been eliminated in the revised paper. The paper Bakker et al. (2020) has been added as a reference here and the content has been updated based on this paper in the revision (Line 47-49). This paper is the most recent published compilation of measured $pCO_2$.

57: References needed for RS $pCO_2$ here.

Response: The suggestion has been taken in the revision. Bai et al. (2015) has been added here in the revised paper (Line 68).

Figure 1. What is meant by "standard deviations"? Standard deviations of grid values, or deviations of values within each grid cell?

Response: It is the temporal standard deviation of the RS $pCO_2$ values on each grid box. In the revision, this explanation has been added (Line 80).

Figure 2. Consider changing this map to a 2 dimensional histogram showing number summers with measurements (probably with colored bins).

Response: This map has been changed in the revision showing the number of summers with underway data (Figure 2).

109: These estimates were... change "data" to "estimates" in this section since $pCO_2$ is not measured. In figure 1 as well.

Response: The changes have been made here (Line 133, 135) and in Figure 1 in the revised paper.

138: Where is this symbol used?

Response: The symbol $\langle \cdot \rangle$ is only used once for expected value in Eq. (2) in the paper. In the revised version, we have replaced the symbol $\langle \cdot \rangle$ by $E[\ ]$ (Line 171), which is more commonly used in statistics and science, while $\langle \cdot \rangle$ is a symbol commonly used in the field of theoretical physics.

141: how many EOFs were used? Say here.

Response: The suggestion has been taken in the revised paper (Line 175-176). Eight EOFs were used for every year except 2000, which had only four EOFs because the year had only five grid boxes with observed underway data.

182: This only shows the fields. One must compare this field to other figures to get an indication of how well the reconstruction performs. A plot showing differences between observed and reconstructed values is required.

Response: We have taken your suggestion and have produced two figures showing differences

between observed and reconstructed values in the revision (Figures 9 and 10).

187: It is not enough to say "we fit the model to the data, so it fits the data." Statistics of goodness-of-fit should be presented. Furthermore, demonstrating that the method works requires withholding several cruises worth of data from the training data set and then using those cruises to verify that the method reconstructs the withheld data. Statistics and plots are required to quantify how well the reconstruction does.

Response: Again, this is a cross-validation issue discussed earlier. We have included our results of cross-validation and uncertainty quantification in the revised paper (Line 246-283, Figures 9 and 10, Table 2) as shown in our response to the previous comments.

195: What is meant by reasonable?

Response: In the revised paper the RMSE, 7.3 $\mu$atm, has been provided here (Line 227), which indicates that the reconstruction appears reasonable.

205: it is unclear what is meant by "the large spatial gradient of in situ data."

Response: Here it means the large spatial variation of in situ data. The "gradient" is changed to "variation" in the revision (Line 239).

214: 2.383 is given to excessive precision. An attempt should be made to quantify the uncertainty and the data should be reported to the appropriate precision.

Response: The suggestion is taken. The rate has been given as 2.4±0.8 $\mu$atm/yr in the revised paper (Line 287, Figure 11).

229: why would a higher rate be expected in a marginal sea? I would argue that 2.4 uatm/year is completely within expectations of the atmospheric $p$CO$_2$ trend over this time period given the large uncertainties in this approach and the likely natural variability in surface ocean $p$CO$_2$.

Response: Considering the uncertainty in the rate is 0.8 $\mu$atm/yr, we agree with the reviewer that our rate is consistent with the trend shown at Station HOT in the Pacific. In the revised paper, this statement has been included (Line 296) and the sentence about "a higher rate be expected in a marginal sea" has been deleted.

[revised manuscript text omitted]

---

## Author Response (AR2)

Our response is in blue with line numbers listed where revisions are made.

**Report #1**

The paper is much improved and the authors should be commended on their efforts. The new analyses based on the SEATS data are very helpful, and much work has been done to better quantify the uncertainty. It is also excellent that the paper is confirming that it is presenting new data, and this will elevate the impact of the work.

Response: Thank you very much for your positive confirmation of the improvement and value of our paper.

I yet feel the uncertainty assessment would benefit from another revision. It is not enough to omit one 0.5x0.5 degree grid cell at a time and recompute estimates in that grid cell. That is just a test of how well the method works when the pCO2 estimate has proximal (in both space and time) measurements. The test should omit entire cruises or entire swathes of data in the tests (i.e., the western or easternmost halves of all data in each year... though even that would provide somewhat of an underestimate since the estimates would always benefit from temporally-proximal measurements).

Response: Following your suggestion, we have performed another cross-validation analysis: Leaving out the western half and eastern halves of the data in a year. The analysis was done for the years with better spatial coverage: 2007, 2009, and 2012. The resulted RMSE is within the range of the RMSEs of the leave-one-out cross validation. This is another confirmation of the reliability of our reconstruction. The details of this cross-validation calculation are as follows.

The western halves (longitude < 115.5º E) of data in 2007, 2009, and 2012 were removed, respectively. The reconstructions were made using the remaining half data. The resulted RMSE between the removed data and the reconstructed data were 2.77, 4.46, and 3.82 µatm for 2007, 2009, and 2012, respectively. Similarly, when the eastern halves (longitude >115.5º E) of data in these years have been removed, the RMSEs are 4.32, 3.66, and 3.55 in 2007, 2009, and 2012, respectively. These values fall in the range of the RMSEs of the leave-one-out cross validation. In the revision, this leave-half-out cross validation has been included in the revised paper (See Lines 257-266).

This type of cross-validation can have infinitely many kinds of variations, depending on the size of the region where the in situ data are withheld. It is also obvious that the regional leave-out would not work if the remaining half has no in situ data. In this revision, rather than exhausting all possibilities, we have only included the regional

leave-out cross-validation for 2007, 2009, and 2012 according to the 115.5ºE meridional line.

Also, I urge the authors to focus on estimate bias for each year instead of estimate RMSE, and to propagate the mean bias for each year's estimates into an estimate of the error in the average fluxes. Probably the most important quantity is RMSE of the annual average pCO2 (similar to average annual absolute bias).

Response: For the bias estimate, we have used the difference between the reconstructed $pCO_2$ and the observed underway $pCO_2$. Figure 9 in the paper shows the difference for each grid in each year. To show the statistics of this figure, we generated boxplots (Figure R1) of the difference data.

Although the bias propagation is an important question, it is not the focus of this data reconstruction paper. The flux of air-sea $pCO_2$ will be among our further studies. The bias of the reconstructed $pCO_2$ will then be transferred to the flux.

As we pointed out in Section 2.1 in our paper "This study focuses on the summer data since the greatest temporal coverage of the sampling occurs in summer." Data in other seasons are scarce. It is still under investigation whether they can be used for annual reconstruction. When the annual reconstruction is made, the corresponding RMSE will be provided.

[Figure]

Figure R1: The boxplot of the difference between the reconstructed summer $pCO_2$ and the observed underway $pCO_2$ (unit: μatm) in 2000, 2001, 2004–2009, 2012, and 2014–2017. The upper limit is defined as Q3+1.5×(Q3-Q1), where Q3 is the third quartile and Q1 is the first quartile. The lower limit is defined as Q1-1.5×(Q3-Q1). The outliers outside the range determined by the upper and lower limits are not shown.

Their response to "1. The model should not be used in any region where there is no fitting data. This includes most of the South China Sea south of ~12.5 N" did not convince me. The method is only valid when its estimates can be trusted. With no validation measurements in the South China Sea, the results cannot be trusted in that region. The paper would be stronger if the results from that region were omitted from

at least one regional trend, or held aside and given an appropriate caution.

Response: The reconstruction is a spatial prediction of the $p\mathrm{CO}_2$ field for the region without in situ observations. We could do so because we have used the EOFs patterns from remote-sensing derived estimates, anchored on the sparse in situ data. This has been described in detail in the method section 2.3 in the paper.

Ideally, the in situ data should be evenly distributed in the entire South China Sea. Hence, the reconstruction can be validated for the entire region. However, the real situation is not the case. The in situ data are limited to regions as shown in Figure 3 in the paper. Nonetheless, our SOG reconstruction can still be made using the EOF patterns.

I didn't understand the authors' response to "If the Bai et al. approach gives a different average pCO2 than the in situ measurements, then the climatology created from the remote sensing product should not be used to generate the Standardized Anomalies of Obs. Data (as indicated in Figure 1). I believe an independent climatology would then be needed. Otherwise, a significant average bias would have to be compensated by a large average value for one or more EOFs." The response seemed to explain how EOFs work using jargon. Perhaps I just didn't understand, but the reply didn't seem to address the concern. The concern is that a big error in the mean state in the underlying climatology would have to be compensated by spurious scaling of EOFs that are intended to capture anomalies from the climatology (not errors in the mean state). The new analysis suggests that the RS pCO2 is indeed overestimating average pCO2, and sometimes by up to 50 uatm (separately, this should be quantified with an average bias statistic across all years as well). This means the climatological values might indeed be a poor representation of the true state of the region.

Response: Yes, indeed. Our EOF reconstruction step is for the anomalies. The full field is recovered by adding the climatology and then multiplied by standard deviation, as illustrated in Figure 1 and described in Section 2.3 in the paper. We understand that the remote-sensing derived $p\mathrm{CO}_2$ estimates have overestimated the field as shown in Figure 5 in the paper. When the accuracy of the remote-sensing data is improved, it is certainly helpful to improve our reconstruction. However, this RS dataset is the best so far available for computing the spatial pattern. We actually tested ocean model-derived $p\mathrm{CO}_2$. The result was inferior. In addition, our reconstructed results may help correct the bias from the remote-sensing estimates, which needs further studies.

For the RS $p\mathrm{CO}_2$ bias estimate, we have used the difference between the remote-sensing derived $p\mathrm{CO}_2$ and the observed underway $p\mathrm{CO}_2$ for each grid in each year as shown in Figure 5 in the paper. Figure R2 below shows the statistics of Figure 5's data using boxplots.

[Figure]

Figure R2: The boxplot of the difference between the remote-sensing derived summer $pCO_2$ and the observed underway $pCO_2$ (unit: μatm) in 2000, 2001, 2004–2009, 2012, and 2014. The upper limit is defined as Q3+1.5×(Q3-Q1), where Q3 is the third quartile and Q1 is the first quartile. The lower limit is defined as Q1-1.5×(Q3-Q1). The outliers outside the range determined by the upper and lower limits are not shown.